

# Satellite-derived methane hotspot emission estimates using a fast data-driven method

Michael Buchwitz[1], Oliver Schneising[1], Maximilian Reuter[1], Jens Heymann[1], Sven Krautwurst[1], Heinrich Bovensmann[1], John P. Burrows[1], Hartmut Boesch[2,3], Robert J. Parker[2,3], Rob G. Detmers[4], Otto P. Hasekamp[4], Ilse Aben[4], André Butz[5], Christian Frankenberg[6,7]

[1]Institute of Environmental Physics (IUP), University of Bremen, Bremen, Germany
[2]Earth Observation Science, University of Leicester, Leicester, UK
[3]NERC National Centre for Earth Observation, Leicester, UK
[4]SRON Netherlands Institute for Space Research, Utrecht, The Netherlands
[5]Karlsruhe Institute of Technology (KIT), Karlsruhe, Germany
[6]Division of Geological and Planetary Sciences, California Institute of Technology, California, Pasadena, CA, USA
[7]Jet Propulsion Laboratory, Pasadena, CA, USA

*Correspondence to*: Michael Buchwitz (Michael.Buchwitz@iup.physik.uni-bremen.de)

**Abstract.** Methane is an important atmospheric greenhouse gas and an adequate understanding of its emission sources is needed for climate change assessments, predictions and the development and verification of emission mitigation strategies. Satellite retrievals of near-surface-sensitive column-averaged dry-air mole fractions of atmospheric methane, i.e., $XCH_4$, can be used to quantify methane emissions. Here we present a simple and fast method to estimate emissions of methane hotspots from satellite-derived $XCH_4$ maps. We apply this method to an ensemble of $XCH_4$ data products consisting of two products from SCIAMACHY/ENVISAT and two products from TANSO-FTS/GOSAT covering the time period 2003-2014. We obtain annual emissions of the source areas Four Corners in the southwestern USA, for the southern part of Central Valley, California, and for Azerbaijan and Turkmenistan. We find that our estimated emissions are in good agreement with independently derived estimates for Four Corners and Azerbaijan. For the Central Valley and Turkmenistan our estimated annual emissions are higher compared to the EDGAR v4.2 anthropogenic emission inventory. For Turkmenistan we find on average about 50% higher emissions with our annual emission uncertainty estimates overlapping with the EDGAR emissions. For the region around Bakersfield in the Central Valley we find a factor of 6-9 higher emissions compared to EDGAR albeit with large uncertainty. Major methane emission sources in this region are oil/gas and livestock. Our findings corroborate recently published studies based on aircraft and satellite measurements and new bottom-up estimates reporting significantly underestimated methane emissions of oil/gas and/or livestock in this area in inventories.




## 1 Introduction

Methane ($CH_4$) is the second most important human-emitted greenhouse gas - directly after carbon dioxide - and increases in its atmospheric abundance contribute significantly to global warming (IPCC, 2013). Accurate knowledge of its sources and

sinks and the origins of any changes are needed for the accurate prediction of future climate change, the attribution of change, and the development of mitigation strategies, but our current knowledge about the various natural and anthropogenic methane sources and sinks has significant gaps (e.g., Rigby et al., 2008; Dlugokencky et al., 2009; IPCC, 2013; Kirschke et al., 2013; Houweling et al., 2014; Nisbet et al., 2014; Jeong et al., 2014; Alexe et al., 2015; Schaefer et al., 2016; Miller and Michalak et al., 2016).

Near-surface-sensitive satellite observations of atmospheric methane have been used in recent years to obtain quantitative information on methane emissions (e.g., Alexe et al., 2015; Bergamaschi et al., 2007, 2009, 2013; Bloom et al., 2010; Turner et al., 2015, 2016; Fraser et al., 2013; Monteil et al., 2013; Cressot et al., 2014; Wecht et al., 2014a, 2014b; Kort et al., 2014). Nevertheless, there are still many important aspects, which need further investigation. For example, aspects related to

the recent renewed methane growth (e.g., Houweling et al., 2014) need an unambiguous explanation, and better knowledge about aspects related to specific but evolving man-made emission sources (e.g., Schneising et al., 2014) is required.

Several important issues for the future management and mitigation of methane emissions have not yet been addressed adequately such as verification of emission inventories and reported emissions per region (country down to city scale) (e.g.,

Ciais et al., 2014). The latter aspect has been addressed for future satellite missions, especially for $CO_2$ in the context of the proposed CarbonSat mission (Bovensmann et al., 2010; Velazco et al., 2011; Buchwitz et al., 2013; Pillai et al., 2016) using performance assessments based on simulated satellite observations (ESA, 2015) but so far only few studies have been published using real satellite data (e.g., Wecht et al., 2014a; Turner et al., 2015, 2016, for USA methane emissions). Here we report on an attempt to use satellite methane retrievals to estimate the methane emissions of Azerbaijan and Turkmenistan,

which are both important oil and gas producing countries, and also apply our method to two regions in the USA. All four studied regions show methane enhancements relative to their surrounding area in satellite-derived $XCH_4$ maps.

This manuscript is structured as follows: In Sect. 2 we introduce briefly the satellite data which have been used in this study. In Sect. 3 we describe the analysis method developed to derive methane emissions of (relatively) well localized emission hot

spots from satellite $XCH_4$ retrievals. The results are presented and discussed in Sect. 4 and a summary and conclusions are given in Sect. 5.



## 2 Satellite data

During recent years the retrieval of near-surface-sensitive column-averaged dry-air mole fractions of atmospheric methane
($CH_4$) and carbon dioxide ($CO_2$), i.e., $XCH_4$ and $XCO_2$, from the satellite sensors SCIAMACHY (Burrows et al., 1995; Bovensmann et al., 1999) onboard ENVISAT and TANSO-FTS onboard GOSAT (Kuze et al., 2009, 2016) significantly evolved and improved (e.g., Buchwitz et al., 2015, 2016a, 2016b; Butz et al., 2011; Dils et al., 2014; Frankenberg et al., 2011; Parker et al., 2011, 2015; Schneising et al., 2011, 2012, 2014; Yoshida et al., 2013).

For this study we use the latest data sets of $XCH_4$ retrievals from SCIAMACHY and GOSAT as generated by different research teams of the GHG-CCI project (Buchwitz et al., 2015) of ESA's Climate Change Initiative (CCI, Hollmann et al., 2013). The four satellite $XCH_4$ products used for this study are publicly available and have been obtained from the GHG-CCI website (http://www.esa-ghg-cci.org), where also detailed documentation is available (e.g., Algorithm Theoretical Basis Documents (ATBDs), Comprehensive Error Characterization Reports (CECRs), Product Validation and Intercomparison
Report (PVIR, Buchwitz et al., 2016a)).

Table 1 presents an overview about the four $XCH_4$ satellite data products used in this study. As can be seen, these comprise two SCIAMACHY $XCH_4$ data products retrieved with the WFMD (Buchwitz et al., 2000; Schneising et al., 2011, 2012, 2013) and IMAP (Frankenberg et al., 205, 2006, 2008a, 2008b, 2011) retrieval algorithms, i.e., the GHG-CCI products
CH4_SCI_WFMD and CH4_SCI_IMAP. In addition, we use the two GOSAT products CH4_GOS_OCPR (Parker et al., 2011, 2015) and CH4_GOS_SRFP (Butz et al., 2011, 2012). The $XCH_4$ "full physics" (FP) retrieval algorithm used to generate the latter product is also known as "RemoTeC" and the algorithm to generate the CH4_GOS_OCPR product is the University of Leicester $XCH_4$ "$CO_2$ proxy" (PR) algorithm. The two SCIAMACHY $XCH_4$ algorithms are also "proxy" algorithms. Here, the $XCH_4$ product is obtained by computing the ratio of the retrieved methane column and the
simultaneously retrieved $CO_2$ column multiplied by a correction factor for $XCO_2$ variations using a $CO_2$ model (Frankenberg et al., 2005). The FP algorithm does not require this $CO_2$ correction as $XCH_4$ is retrieved directly, which is an advantage compared to PR algorithms. However, each algorithm has different advantages and disadvantages. An advantage of the $XCH_4$ PR algorithms is that atmospheric light path related errors due to not perfectly considered wavelength dependent scattering by aerosols and clouds largely cancel in the $CH_4$ to $CO_2$ column ratio. This source of error is consequently less of
a problem for PR algorithms compared to FP algorithms, which require more complex radiative transfer modelling and stricter quality filtering compared to the PR products (see also Schepers et al., 2012, for PR and FP algorithms and





corresponding data products but also Buchwitz et al., 2015, 2016a, 2016b). As a consequence, FP data products are typically much sparser compared to PR products, but are independent of the $CO_2$ model used.

The latest validation results for the GHG-CCI $XCH_4$ data products are presented and discussed in Buchwitz et al., 2016a.
These were obtained by comparison of the satellite retrievals with ground-based $XCH_4$ observations of the Total Carbon Column Observing Network, TCCON (Wunch et al., 2011, 2015). As shown in Buchwitz et al., 2016a, the GOSAT $XCH_4$ products are very stable, i.e., do not show any significant trend of the difference with respect to TCCON. For SCIAMACHY the situation is more complex due to detector problems in later years resulting in larger noise but also bias issues. For example, as shown in Buchwitz et al., 2016a, the IMAP product suffers from a bias (a discontinuity in $XCH_4$) in 2010. For
this reason, we decided to restrict the use of the SCIAMACHY products in this study to the period 2003 – 2009. The achieved single measurement precision (random error) for SCIAMACHY $XCH_4$ is in the range 30-80 ppb (2-5%) depending on time period and product and approximately 16 ppb (~1%) for GOSAT. Systematic errors ("relative accuracy" or "relative bias") are around 10-15 ppb (~0.6%) for SCIAMACHY and approximately 6 ppb (~0.3%) for GOSAT.

Annual average composite maps of the four data products are shown in Figs. 1 and 2. Figure 1 shows year 2004 SCIAMACHY $XCH_4$ at $0.5^o$ x $0.5^o$ resolution as retrieved using the WFM-DOAS (WFMD) algorithm (Schneising et al., 2011). Also shown are zooms for the three target regions investigated in this study. Figure 2 shows year 2004 SCIAMACHY IMAP-DOAS (IMAP) $XCH_4$ and year 2010 $XCH_4$ as retrieved using the two GOSAT algorithms. As can be seen, the spatial coverage of the GOSAT products is quite sparse. A single GOSAT observation requires more time (4 seconds) compared to
a SCIAMACHY observation (typically 0.25 seconds for the spectral regions relevant for this study) and, therefore, GOSAT provides less observations in a given time period than SCIAMACHY. On the other hand, the GOSAT ground pixel size is smaller (10 km diameter) compared to SCIAMACHY (approximately 30 km along track times 60 km across track), which results in a higher fraction of cloud free observations for GOSAT. Furthermore, SCIAMACHY is in nadir (downlooking) observation mode only about 50% of the time. Overall the total number of quality filtered observations as contained in the
data products is larger for SCIAMACHY compared to GOSAT. Furthermore, the spatial sampling of GOSAT comprises non-contiguous ground pixels, which results in large data gaps (even in yearly averages). Consequently, GOSAT is typically (i.e., in normal observation mode) not optimal for small-scale hotspot applications but as shown in this manuscript, GOSAT provides results for the selected source regions which agree reasonably well with the results obtained using SCIAMACHY. In the remainder of this manuscript we focus on obtaining methane emission estimates for the source areas shown in Fig. 1.





## 3 Analysis method

In this section we describe the analysis method used to obtain methane emission estimates for source regions such as those
shown in Fig. 1, i.e., for regions showing elevated methane relative to their surrounding area in time-averaged satellite-derived $XCH_4$ maps.

The satellite $XCH_4$ input data used in this study are the GHG-CCI Level 2 (i.e., individual ground-pixel observations) data products as described in the previous section (see also Tab. 1). The first step in the analysis comprises gridding (averaging) these products using a regular latitude/longitude grid (here: $0.5^o \times 0.5^o$) to obtain maps of annual averages (see Figs. 1 and 2). These mapped $XCH_4$ products are then used in this study for further analysis.

The second step comprises the definition of a source region and a surrounding (or background) region. The latter is an extended region surrounding the source region (specific examples are shown in Sect. 4).

The third step comprises the determination of the methane enhancement over the source region relative to its surrounding area, $\Delta XCH_4$. This methane enhancement is computed by subtracting the mean value of $XCH_4$ in the surrounding region from the mean $XCH_4$ value over the source region.

To reduce potential effects related to a location dependent weighting of tropospheric and stratospheric contributions on $XCH_4$ (note that mean stratospheric $CH_4$ mixing ratios are typically lower compared to tropospheric mixing ratios) we apply a correction called "elevation correction" (EC) similar as also described in Kort et al., 2014 (and implicitly also applied in Schneising et al., 2014). The purpose is to correct for $XCH_4$ variations due to variations of surface elevation/pressure and tropospause height. The corrected $XCH_4$ is obtained from the original $XCH_4$ by adding 7 ppb per 1 km surface elevation increase relative to mean sea level. For surface elevation we use a surface elevation map (also $0.5^o \times 0.5^o$) based on the GTOPO30 Digital Elevation Model (DEM) (obtained from https://lta.cr.usgs.gov/GTOPO30). The value of 7 ppb/km has been obtained by fitting a linear function to pairs of uncorrected original $XCH_4$ and corresponding surface elevation. We found that the exact value depends somewhat on region, time period and satellite data product but is typically within 7 +/- 2 ppb/km. We found that applying EC typically results in similar or somewhat lower emission estimates compared to inversions where this correction is not applied.



The fourth step comprises the conversion of the methane enhancement over the source region, $\Delta XCH_4$, to a source region emission estimate ($E_e$; unit: $MtCH_4$/year = $TgCH_4$/year) using conversion factor CF:

$$E_e = \Delta XCH_4 \cdot CF. \tag{1}$$

The basic idea is that a relatively well isolated emission source will result in an $XCH_4$ enhancement, $\Delta XCH_4$, in an area at and around the emission hotspot relative to its surrounding, i.e., that there will be a spatial correlation between a local emission and a local $XCH_4$ enhancement (compare also the two maps shown in Fig. 3 top left and top right, which will be discussed in detail below).

The conversion factor CF in Eq. (1) is computed as follows (see Annex A for additional explanations):

$$CF = M \cdot M_{exp} \cdot L \cdot V \cdot C. \tag{2}$$

15 Here M is a constant conversion factor ($5.345 \cdot 10^{-9}$ $MtCH_4$/km$^2$/ppb) needed to convert a methane mole fraction change to a methane mass change per area for standard conditions, i.e., for surface pressure $p_{surf}$ = 1013 hPa. $M_{exp}$ is a dimensionless factor used to approximately correct for the actual mass (mass $M_i$ of the i-th grid cell). It is calculated using the surface elevation map also used for the determination of the elevation correction (EC) as described above:

$$M_{exp} = \frac{<M_i>}{M} \approx \frac{<p_i>}{1013.0} \approx < e^{-z_i/H} >_i. \tag{3}$$

Here $p_i$ is the surface pressure of the i-th grid cell (in hPa) and $z_i$ is the surface elevation of the i-th grid cell (in km), H is the assumed scale height (8.5 km) and $<\cdot>$ and $<\cdot>_i$ denotes averaging over all grid cells of the source region. As shown below, the uncertainty of our method is not dominated by the approximation used to compute $M_{exp}$ (namely the use of surface

25 pressure or elevation rather than actual mass).

The dimension of the remaining factor ($L \cdot V \cdot C$) is km$^2$/year, i.e., area divided by time or length times velocity and can be interpreted as the effective methane emission accumulation time of air parcels travelling over the source region area or the effective velocity V of air parcels travelling an effective length L over the source region. Here we use the latter

30 interpretation, i.e., L is length (in km) and V is velocity (in km/year). We compute L as the square root of the (pre-defined) source area.



Factor C is dimensionless and in this study we use C = 2.0. This choice is motivated using the simple model of an air parcel travelling with constant horizontal wind speed V over a homogeneous source region of length L accumulating methane during an accumulation time L/V (see Annex A). When leaving the source area, the methane enhancement of the air parcel, i.e., the concentration difference after and before entering the source region, is twice the mean methane enhancement over

the source region due to the assumed linear increase of the methane enhancement of the air parcel when travelling over the source region (see Annex A). Our method basically assumes that the emission of the source region only results in a $XCH_4$ enhancement over the source region. However, also the surrounding area may contain elevated $XCH_4$ from sources located in the surrounding area or from methane inflow from other regions into the surrounding area (including the source region). As our method neglects this, our method tends to underestimate the emission of the source region. As explained below, we

aim at quantifying the impact of the choice of the surrounding region by varying its size and shape.

The value of V has been obtained by "calibrating" our method using global methane data sets obtained from the Copernicus Atmosphere Monitoring Service (CAMS, https://atmosphere.copernicus.eu/). Specifically, we use CAMS methane emissions and atmospheric methane version v10-S1NOAA as generated via the TM5-4DVAR assimilation system assimilating

National Oceanic and Atmospheric Administration (NOAA) $CH_4$ surface observations (an earlier version of this method and resulting data products is described in Bergamaschi et al., 2009). Based on this data set we computed annual emissions and corresponding annual $XCH_4$ at the original CAMS data set resolution of $6^o$ longitude times $4^o$ latitude. The corresponding maps for the year 2003 are shown in Fig. 3 (top row).

The CAMS year 2003 $XCH_4$ map shown in Fig. 3 top left has been used to derive methane emissions using Eq. (1) and varying parameter V (the only free parameter of our model) until the mean difference between our estimated emissions and the "true" CAMS emissions is zero. We found that this is the case for V = 1.1 m/s (converted to km/year). The resulting map of retrieved emissions is shown in Fig. 3 bottom right. This map has been obtained using an automatic procedure: For all CAMS $6^o x 4^o$ grid cells (except for the ones at the border) the $XCH_4$ value of this grid cell has been obtained and is

interpreted as a potential source region value. The neighboring cells define the surrounding (background) of the potential source region and its $XCH_4$ mean value and standard deviation has been computed. A methane enhancement, $\Delta XCH_4$, has been computed as "source minus background value" as described above. If the resulting $\Delta XCH_4$ value is larger than 0.5 times the standard deviation of the $XCH_4$ values in the surrounding, then the corresponding cell is flagged as a methane "hotspot cell" and its $\Delta XCH_4$ value is converted to an emission using the approach described above (Eq. (1)). The

corresponding results are shown as map in Fig. 3 bottom right and can be compared with the "true" emission map shown in Fig. 3 top right. As can be seen in Fig. 3, N = 125 hotspot cells have been found using the described procedure.

Figure 3 bottom left shows x-y plots of estimated emissions versus "true" (i.e., CAMS) emissions (top) and estimated minus true emissions versus true emissions (bottom). The mean difference "estimated-true" is 0.00 MtCH$_4$/year (this must be the





case as V = 1.1 m/s has been determined by minimizing this difference). The standard deviation of the difference is 0.59 $MtCH_4$/year, the linear correlation coefficient R is 0.81 and the red line shows the resulting line from a linear fit. As can be seen, the (red) line originating from the linear fit has a positive slope but does not perfectly agree with the (green) 1:1 line (our single parameter model does not permit to also optimize the slope of the fitted line).

Figure 4 is similar as Fig. 3 but shows results for the year 2012. Here the difference "estimated-true" is not exactly zero but 0.01 MtCH4/year. In contrast to Fig. 3, V has not been fitted. Instead, the pre-defined value of V = 1.1 m/s has been used. Figure 4 shows very similar "estimated-true" differences compared to Fig. 3. This demonstrates that the effective wind speed V as obtained from year 2003 data is valid also for other years.

The results shown in Figs. 3 and 4 are combined in the single Fig. 5. As can be seen from Fig. 5 (top), the overall correlation of the retrieved and true emissions is 0.81, the mean difference (estimated minus true) is 0.00 $MtCH_4$/year and the standard deviation of the difference is 0.53 $MtCH_4$/year. As explained, these results have been obtained using constant values for fit parameter V (= 1.1 m/s) and correction factor C (= 2.0) (Eq. (2)). Several attempts have been undertaken in order to find out

15  if the use of regionally and/or time dependent C values can reduce the difference of the estimated and the true methane emission, however (so far) without success. For example, it has been investigated if the emission difference is correlated with mean wind speed (using ECMWF ERA Interim from www.ecmwf.int/, Dee et al., 2011) but no significant correlation between emission error and mean wind has been found. The time dependence of the estimated emission, $E_e$, is therefore nearly entirely driven by the satellite-derived methane enhancement, $\Delta XCH_4$.

Finally, the (1-sigma) uncertainty of $E_e$ has been estimated. This has been done as follows: Figure 5 also shows the emission difference (estimated minus true; see middle and bottom panels) as a function of the estimated emission. Figure 5 middle also shows (in red) the corresponding mean values (crosses) and standard deviations (vertical bars) for several emission bins (non-equidistant to ensure a sufficiently large number of data points within each bin). Also shown in Fig. 5 (middle and

25  bottom) are dotted red lines computed as $f(E_e) = 0.3 + 0.5 \cdot E_e$. This function and its parameters has been chosen such that the red vertical bars (1-sigma range) are located within the range defined by $f(E_e)$, i.e., most of the emission differences are located within +/- $f(E_e)$ (Fig. 5 middle). Therefore, $f(E_e)$ is a reasonable description of the 1-sigma uncertainty of the estimated emissions. Based on this it is concluded that the 1-sigma uncertainty of the estimated emission due to uncertainty of the overall conversion factor (CF) can be well described using this formula:

$$\sigma_{CF} = 0.3 + 0.5 \cdot E_e. \tag{4}$$

Here the units of $\sigma_{CF}$ and $E_e$ are $MtCH_4$/year. The total uncertainty, $\sigma_{tot}$, consists of the uncertainty of the conversion factor, $\sigma_{CF}$, and the uncertainty of the obtained methane enhancement, $\sigma_{\Delta XCH4}$, as obtained from the satellite data (see Eq. (1)). The





latter is assumed to be dominated by methane variations in the surrounding area (primarily because the surrounding region may contain regions of elevated methane due to sources located outside the target region). This contribution to the total uncertainty is estimated by varying the size of the surrounding region (see following section). The total uncertainty is computed as follows:

$$\sigma_{tot} = \sqrt{\sigma_{\Delta XCH4}^2 + \sigma_{CF}^2} \qquad (5)$$

The method described in this section has been applied to the described SCIAMACHY and GOSAT $XCH_4$ data products and for each of the pre-defined source regions annual average emissions and their uncertainties have been obtained for all

products. The results are presented in the following section.

## 4 Results and discussion

In this section we present the results from applying the methane emission inversion method described in the previous section to obtain emission estimates for four areas: the Four Corners area in the south-western USA (Sect. 4.1), in the southern part of the Central Valley in California (Sect. 4.2) and the two countries Azerbaijan and Turkmenistan (Sect. 4.3). All these areas show elevated methane relative to their surrounding (Fig. 1). The spatial locations of these areas as well as key parameters used to convert the observed methane enhancements to annual methane emissions are listed in Tab. 2.

### 4.1 Four Corners area, USA

Four Corners is a region in the USA named after the quadripoint where the boundaries of the four states Utah, Colorado, Arizona and New Mexico meet. The Four Corners area is one of the largest methane hotspots in the USA (Kort et al., 2014;

Wecht et al., 2014b; Frankenberg et al., 2016). The San Juan Basin, located in the Four Corners area, is a geologic structural basin and primarily a natural gas production area, mostly from coal bed methane and shale formations (e.g., Frankenberg et al., 2016, and references given therein). Figure 6 shows annually averaged $XCH_4$ from the four satellite $XCH_4$ products as used in this study at and around Four Corners. Here the $XCH_4$ is shown as anomaly, to be able to better compare the spatial pattern of the shown products. As can be seen, all satellite products show that $XCH_4$ is enhanced in the Four Corners area

relative to the surrounding area (for the OCPR product this is difficult to see because the obtained enhancement is the smallest of all products). Figure 6 shows the chosen source region as (inner) rectangle. The outer rectangle (last column and last row) shows the "default" surrounding area. As described above, the methane enhancement $\Delta XCH_4$ is computed as the difference between the $XCH_4$ mean value in the source region minus the $XCH_4$ mean value in the surrounding region. For the inversion the size of the surrounding area is varied to determine the sensitivity of the computed $\Delta XCH_4$ with respect to





the chosen background region. For this purpose the latitudes and longitudes of the rectangular box, which defines the surrounding area, are varied by adding all combinations of $0^o$, $1^o$, $2^o$, and $3^o$ in the latitude and longitude directions. The standard deviation of the resulting $\Delta XCH_4$ is used as an estimate of $\sigma_{\Delta XCH4}$ (see Eq. (5)).

Figure 7 shows the resulting $XCH_4$ enhancements for all years and all satellite data products including (1-sigma) uncertainty estimates (i.e., $\sigma_{\Delta XCH4}$) as vertical bars. As can be seen, all $\Delta XCH_4$ values are positive. This shows that a positive Four Corners methane enhancement is present for all years in all satellite products. The methane enhancement is on average about 10 ppb but shows significant variation depending on satellite product and year.

These methane enhancements and their uncertainties are converted to Four Corners area annual methane emissions using the method described in Sect. 3. The results are shown in Fig. 8. The estimated emissions are in the range 0.42 – 0.57 $MtCH_4$/year (range of annual mean values of the four satellite products). Taking into account the (large) uncertainty of the estimated annual emissions, this is in good agreement with published values as shown in Fig. 8. For example, Kort et al., 2014, report 0.59 $MtCH_4$/year for the time period 2003-2009 (based on SCIAMACHY and ground-based Fourier-Transform
(FT) spectrometer observations) and Turner et al., 2015, report the range of 0.45 -1.39 $MtCH_4$/year for the time period 2009-2011 (based on an analysis of GOSAT data). The good agreement with the published values indicates that the method used here appears to be capable to deliver reasonable emission estimates even if the source area is much smaller than the $6^o x 4^o$ regions used for calibrating our inversion method. The agreement is surprisingly good given the large (1-sigma) uncertainty values shown in Fig. 8 (approx. 0.6 $MtCH_4$/year (~100%) and dominated by $\sigma_{CF}$ as can be concluded from a comparison with
$\sigma_{\Delta XCH4}$ shown in Fig. 7 (~20%)). Our reported uncertainty of the annual averages seems to be too conservative (at least for quantifying the Four Corners area emissions).

Figure 8 also shows the total anthropogenic emissions during 2003-2008 as obtained from the EDGAR v4.2 data base (obtained from http://edgar.jrc.ec.europa.eu/part_CH4.php) for the Four Corners source region. The mean value of the
annual EDGAR emissions is 0.17 $MtCH_4$/year. As can be seen, the EDGAR emissions are too low by approximately a factor of three.

## 4.2 Central Valley, California, USA

California emits large amounts of methane, approximately 2-3 $MtCH_4$/year (Turner et al., 2015) and major emission sources are livestock, gas/oil and landfills/wastewater (e.g., Wecht et al., 2014b).  According to the EDGAR v4.2 emission data base total anthropogenic methane emissions are largest around Los Angeles and San Francisco dominated by landfill/wastewater and gas/oil related emissions and in the area in between, in the Central Valley, emissions are dominated by livestock emissions (see Wecht et al., 2014b, their Fig. 1).



The Central Valley in California shows up as a methane hotspot in satellite data (see Fig. 9) with largest values in the southern part of the Central Valley around Bakersfield, an important oil and gas producing area (e.g., Jeong et al., 2014; Guha et al., 2015) and an area with significant methane emissions from dairy and livestock (e.g,, Wecht et al., 2014b; Guha et al., 2015), extending up to the city of Fresno or even further towards Modesto / San Francisco. This southern part of the Central Valley is the San Joaquin Valley. In this study we define Central Valley as the rectangular region specified by the latitude/longitude range as listed in Tab. 2, corresponding to the region where the satellite $XCH_4$ is highest. This region roughly corresponds to the San Joaquin Valley. According to EDGAR this region is dominated by livestock methane emissions with significant contributions from gas/oil and landfill/wastewater related emissions.

Figure 9 shows SCIAMACHY WFMD (and IMAP) $XCH_4$ for year 2004 over California and also shows the Central Valley source region as defined for this study (inner rectangle of Fig. 9 top left) and its "default" surrounding area (outer rectangle Fig. 9 top right). Figure 9 also shows EDGAR v4.2 total anthropogenic methane emissions for the year 2004 regridded to $0.5^o \times 0.5^o$. As can be seen, the spatial pattern of the EDGAR emissions significantly deviates from the spatial pattern of the satellite $XCH_4$. Whereas in EDGAR the highest values are around San Francisco and around Los Angeles, the satellite-derived atmospheric methane is highest in the area in between, in the Central Valley, particularly in the area around Bakersfield. Methane emissions in the Bakersfield region are supposed to be dominated by dairy and livestock operations (Guha et al., 2015, and references given therein).

For comparison with the satellite data and the EDGAR emissions also the CAMS emissions are shown (Fig. 9 bottom row). On the left (Fig. 9e) the CAMS v10-S1NOAA product is shown, which is based on the assimilation of NOAA methane observations and on the right product v10-S1SCIA (Fig. 9f) based on the additional assimilation of SCIAMACHY IMAP $XCH_4$. Surprisingly, the assimilation of SCIAMACHY $XCH_4$ reduces the derived methane emissions in this region. That the Central Valley SCIAMACHY $XCH_4$ enhancement is not modelled well with optimized emissions obtained from assimilating SCIAMACHY data using the global TM5-4DVAR system is also clearly visible in Bergamaschi et. al., 2009 (their Fig. 2), discussing an earlier (pre-CAMS) version of this data set. As already mentioned, the emissions of California are expected to be in the range 2-3 $MtCH_4$/year (see Turner et al., 2015, their Fig. 6), i.e., larger than the v10-S1NOAA (Fig. 9e) and v10-S1SCIA (Fig. 9f) products suggests. The exact reason why the assimilation of the SCIAMACHY data does not lead to larger estimated emissions in this region is unclear but very likely this is due to the fact that the CAMS inversion system is a global system at quite low spatial resolution and therefore not necessarily optimal for proving reliable emission estimates for regions which are smaller or just on the order of the size of the $6^o \times 4^o$ grid cells shown in Fig. 9 bottom.

As can be seen from Fig. 10, we obtain mean annual emissions in the range 1.05-1.55 $MtCH_4$/year, depending on data product. The estimated uncertainty of the annual emissions is ~1 $MtCH_4$/year (1-sigma) and the inter-annual variations are



20-50% (1-sigma) of the mean emissions, depending on product. Our annual emission estimates are quite uncertain with mean values much higher compared to the emissions as given in the EDGAR v4.2 anthropogenic methane emission inventory. According to EDGAR the total anthropogenic methane emissions in the selected source area are around 0.17 $MtCH_4$/year, i.e., a factor of 6-9 lower than our annual mean estimates. This is unlikely due to the fact that our emissions are

total emissions whereas EDGAR only reports anthropogenic emissions as the fraction of natural methane emissions in California is estimated to be only approximately 3% percent (Wecht et al., 2014b). Our results are broadly consistent with recently published results from CalNex campaign (May – June 2010) aircraft observations (Wecht et al., 2014b) also showing high atmospheric methane concentrations over the southern Central Valley compared to the rest of California and concluding that EDGAR emissions in this region need to be scaled with factors up to around five (see their Fig. 2). Wecht et

al., 2014a, also derived emissions in this area using SCIAMACHY IMAP retrievals. They report that their derived emissions are consistent with the ones presented in Wecht et al., 2014b, and for the Central Valley they found that the derived emissions are a factor of 2-4 higher compared to EDGAR v4.2 (note that their definition of Central Valley is not identical with our definition, which is restricted to the southern part of the Central Valley). They conclude that the livestock emissions in EDGAR are significantly underestimated.

Jeong et al., 2013, present an analysis of methane emissions using atmospheric observations from five sites in California's Central Valley across different seasons (September 2010 to June 2011). They obtained spatially resolved (13 sub-regions) top-down estimates of California's $CH_4$ emissions using in-situ tower data. They report for their region R12, which is similar to but not exactly identical with the area chosen in our assessment, emissions of 0.85 and 0.94 $MtCH_4$/yr (depending on *a priori* assumptions) based on inversion of in-situ tower data (see their Tab. 5 reporting methane emissions in $TgCO_2eq$

computed assuming a global warming potential of 21 $gCO_2eqCH_4/gCH_4$), which is a factor of 3.6 (= 17.89 / 5.01, see their Tab. 5) higher than EDGAR v4.2.

Jeong et al., 2014, also studied this region and presented a new spatially resolved bottom-up inventory of methane for 2010 focusing on methane emissions from petroleum production and natural gas systems in California. They showed that the region around Bakersfield is a major oil and gas production and transmission region in California (see their Fig. 1) and they

found that their emission estimates are 3-7 times higher for the petroleum and gas production sectors compared to official California bottom-up inventories.

Our results corroborate the findings of these independent studies that inventory emissions are underestimated in this region. However we acknowledge the large uncertainty of our estimated annual emissions and cannot rule out that our emission estimates are overestimated, e.g., due to possible methane accumulation in the southern part of the Central Valley.





### 4.3 Azerbaijan and Turkmenistan

Azerbaijan and Turkmenistan are located next to the Caspian Sea (to the west and to the east, respectively) and both
countries are important oil and gas producers. Azerbaijan and Turkmenistan are clearly visible as methane emission hotspots in satellite $XCH_4$ data sets (Fig. 1, Fig. 11).

Figure 11 shows SCIAMACHY WFMD year 2004 $XCH_4$ in the Azerbaijan / Turkmenistan area and emission data base results from EDGAR v4.2 (Fig. 11, bottom left), CAMS v10-S1NOAA (Fig. 11, bottom middle) and CAMS v10-S1SCIA
(Fig. 11, bottom right). In contrast to the results discussed in the previous section, the assimilation of SCIAMACHY data in the TM5-4DVAR assimilation system enhances the emissions around Azerbaijan / Turkmenistan (compare Fig. 11 bottom middle with bottom right).

Figure 12 shows Azerbaijan methane emissions as obtained with our inversion method compared to EDGAR v4.2 emissions.
As can be seen, the satellite-derived emissions are consistent with EDGAR. Note that the CH4_GOS_SRFP product is not shown. Due to the sparse spatial sampling of this product the inter-annual variability is dominated by year-to-year sampling differences. Azerbaijan is surrounded by many other methane emission areas and, therefore, not a well-isolated emission hotspot, i.e., not ideal for our inversion method. The impact of this is largest for the CH4_GOS_SRFP product, which is a sparse data set as the underlying "full physics" retrieval algorithm requires strict quality filtering.

Turkmenistan is much larger in size compared to Azerbaijan (see Fig. 11) but also not a well-isolated emission hotspot. The results for Turkmenistan are shown in Fig. 13. Here the mean values of all estimated emissions are positive (in contrast to Azerbaijan) indicating that the methane concentration over Turkmenistan is higher than its surrounding for all years and all four satellite products. The mean values of the derived emissions are in the range $1.85 - 2.08$ $MtCH_4$/year, which is about
50% larger compared to EDGAR (1.33 $MtCH_4$/year). This may be due to an underestimation of Turkmenistan's oil and gas related methane emissions in EDGAR but one also has to note the large uncertainty of our satellite-derived annual emissions. Furthermore, Turkmenistan is not an ideally-isolated methane hotspot, although the Azerbaijan results do not indicate that this is necessarily a significant issue. Note also that mountains are located southward and eastward of Turkmenistan and this may contribute to a local accumulation (trapping) of atmospheric methane (resulting in an overestimation of our estimated
emissions) and may explain why the elevated methane over Turkmenistan as shown in Fig. 11 is well correlated with the country boundaries. Clearly, more studies are needed to clarify this but this likely requires much more complex inversion methods than the one used in this study (e.g., similar to those presented in Wecht et al., 2014a, and Gentner et al., 2014).



## 5 Summary and conclusions

We have presented a simple but fast method to estimate methane surface emissions of areas showing elevated atmospheric methane concentrations relative to their surrounding area ("methane hotspots") in satellite-derived $XCH_4$ maps, especially in

those derived from SCIAMACHY/ENVISAT. The described "inversion method" is applicable to time-averaged $XCH_4$ data sets (as complex variations due to varying meteorological conditions cannot be considered by our method) and we focus on annual $XCH_4$ maps to derive annual emissions. The method is based on a direct conversion of a localized methane enhancement (relative to its surrounding) using a conversion factor, which mainly depends on the size of the region of interest. The method is calibrated using (low resolution) 2-dimensional methane emission maps and corresponding 2-

dimensional $XCH_4$ maps generated from Copernicus Atmospheric Monitoring Service (CAMS) 3-dimensional atmospheric methane fields. A limitation of our method is its quite large uncertainty. We estimate that the uncertainty of the method is about 80% for annual emissions around 1 $MtCH_4$/year but having better relative uncertainty for larger emissions (down to about 50% for large emissions).

We applied this method to an ensemble of satellite $XCH_4$ data products using two products from SCIAMACHY/ENVISAT and two products from TANSO-FTS/GOSAT as made available via the GHG-CCI project website (http://www.esa-ghg-cci.org/) of ESA's Climate Change Initiative (CCI). These products cover the time period 2003-2014.

The inversion method has been applied to four source areas. Two of the source areas are located in the USA (the Four

Corners area located in the southwestern USA and the southern part of the Central Valley, i.e., the region around Bakersfield and Fresno, in California) and the two other source regions are Azerbaijan and Turkmenistan, which are both important oil and gas producing countries. All four regions clearly show elevated methane relative to their surrounding in satellite-derived $XCH_4$ maps.

For Four Corners we obtain annual emissions in the range $0.42 - 0.57$ $MtCH_4$/year in agreement with published values. For Azerbaijan our estimates are on average close to the total anthropogenic methane emissions of Azerbaijan as given in the EDGARv4.2 (FT2012) emission inventory but for Turkmenistan we obtain about 50% higher emissions on average albeit with large uncertainty. Further study is needed toinvestigate if this is due to an underestimation of Turkmenistan's oil and gas related emissions in EDGAR.

For the region around Bakersfield located in the Central Valley of California, a region of significant oil and gas production and large expected methane emissions from dairy and livestock operations, we obtain mean emissions in the range 1.05-1.55 $MtCH_4$/year, depending on satellite data product. This is about a factor of 6-9 higher than the total methane emissions as given in the EDGAR v4.2 inventory, but of similar magnitude as reported in Jeong et al., 2013, $(0.85 - 0.94$ $MtCH_4$/year)



based on inverse modelling of tower measurements. Our findings also corroborate published results from CalNex campaign aircraft observations during May to June 2010 (Wecht et al., 2014b) showing high methane concentrations over the southern part of the Central Valley, in the San Joaquin Valley, compared to other parts of California and concluding that EDGAR emissions in this area need to be scaled with factors up to around five. They conclude that livestock emissions in EDGAR are significantly underestimated. Another more recent study (Joeng et al., 2014) presented a new bottom-up methane inventory for the year 2010 for California concluding that their emissions are 3-7 times higher compared to official California bottom-up inventories for the petroleum and natural gas production sectors. Nevertheless, our results need to be interpreted with care as the uncertainty of our annual emission estimates is quite large and we cannot rule out that our estimates are somewhat overestimated, e.g., due to possible methane accumulation in the valley.

We recommend further studies to investigate in more detail the reported discrepancy of the satellite-derived emissions with emission inventories in particular for the southern part of the Central Valley in California and for Turkmenistan. We also recommend to use ensembles of satellite products as done in this study in order to determine to what extent key findings are depending on the algorithmic choices which have to be made when developing a retrieval algorithm used to generate a particular $XCH_4$ data product and to what extent the findings depend on the particular satellite instrument used to derive the results. More detailed assessments likely require the use of much more complex approaches compared to the simple method uses in this study. Nevertheless, simple and fast approaches also have a role to play as they permit to perform quick assessments on possible discrepancies with respect to emission inventories or other data sets and can also be used for plausibility checks for more complex approaches.

It is also important to monitor the emissions of major methane source regions in the future. In this context the upcoming satellite mission Sentinel-5-Precursor (S5P) will potentially play an important role. S5P is planned to be launched end of 2016 and will deliver $XCH_4$ at high spatial resolution (7 km at nadir) and with good spatial coverage (2600 km swath width, i.e., daily coverage) (Veefkind et al., 2012; Butz et al., 2012) resulting in methane observations with dense spatio-temporal coverage, which is a significant advantage for methane hotspot detection and related emission quantification compared to the past and present satellites used in this study.

The longer term objective of releasing an observing system comprising instruments having the performance of CarbonSat within a CarbonSat constellation (Bovensmann et al., 2010; Velazco et al., 2011; Buchwitz et al., 2013; Pillai et al., 2016; ESA, 2015) is currently being discussed by the European Space Agency (ESA) and European Union (EU) representatives within the Copernicus program focusing on $CO_2$ (e.g., Ciais et al., 2015). Such a system will provide, when coupled with sparse but accurate ground-based systems, the objective evidence about the global $CH_4$ and $CO_2$ surface fluxes needed for verification and monitoring of emissions and to improve our knowledge on natural carbon fluxes.





**Acknowledgements**

This study has been funded by ESA via the GHG-CCI project of ESA's Climate Change Initiative (CCI). J. H. and R.P. are also funded by an ESA Living Planet Fellowship. The University of Leicester GOSAT retrievals used the ALICE High Performance Computing Facility at the University of Leicester. We thank ESA/DLR for providing us with SCIAMACHY Level 1 data products and JAXA for GOSAT Level 1B data. We also thank ESA for making these GOSAT products available via the ESA Third Party Mission archive. The satellite $XCH_4$ data products have been obtained from the ESA project GHG-CCI website (http://www.esa-ghg-cci.org/) maintained by University of Bremen. Methane emissions and corresponding atmospheric methane fields have been obtained from the Copernicus Atmospheric Monitoring Service (CAMS) website (https://atmosphere.copernicus.eu/; we thank P. Bergamaschi, European Commission Joint Research Centre (EC-JRC), Institute for Environment and Sustainability (IES), Climate Change Unit, Ispra, Italy, for providing us with earlier versions of this data set). We thank the EDGAR team for making available EDGAR anthropogenic methane emission inventory data (obtained from http://edgar.jrc.ec.europa.eu/part_CH4.php). We also thank USGS for making available the GTOPO30 Digital Elevation Model (DEM) data base (obtained from https://lta.cr.usgs.gov/GTOPO30) and ECMWF for meteorological data (obtained from www.ecmwf.int/).

**Annex A: Illustration of emission estimation method**

Figure A1 illustrates the basic idea of the methane emission estimation method explained in Sect. 3. In particular it is illustrated how the observed methane enhancement over the source region (region A in Fig. A1 (a)), $\Delta XCH_4$, is related to the source region emission (E, in mass per time), wind speed magnitude V, and length of the source region. The source region shown here is a rectangle of area $A = L_x L_y$, where wind speed is in the x-direction. Note that length L as given in Eq. (2) corresponds to length $L_y$ of Fig. A1.

The computation of the methane mole fraction enhancement over the source region relative to its surrounding, $\Delta XCH_4$, is computed (see Sect. 3 and Sect. 4) by subtracting the mean value of $XCH_4$ in the surrounding region (region B in Fig. A1 (a)) by the mean value of $XCH_4$ over the source region (region A in Fig. A1 (a)) assuming that the surrounding region does not contain any (significant) emission sources and neglecting atmospheric methane enhancements in the surrounding due to outflow from the source region into the surrounding region (region C in Fig. A1 (a); note that this requires that region B is much larger than region C). As a consequence, the computed mean value of $XCH_4$ in the surrounding is typically overestimated and, therefore $\Delta XCH_4$ and the computed methane emission is too low, i.e., the estimated emission is (typically) a conservative estimate.



Note that the method described in Sect. 3 and used in Sect. 4 is only applied to time averages of atmospheric $CH_4$ to obtain time averaged emissions. This typically means that meteorological situations vary significantly during the selected time period (including large wind speed and wind direction variations) so that detailed structures of the atmospheric methane emission "plumes" originating from local emission sources largely average out resulting in enhanced atmospheric methane over the source region. Note that Fig. A1 (b) only illustrates a "snapshot" in time but not the average over a range of wind speeds and wind directions (assumed to be reasonably well approximated by Fig. A1 (a)).

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



| Product | Sensor | Algorithm | Version | Institute | References |
|---|---|---|---|---|---|
| CH4_SCI_WFMD | SCIAMACHY on ENVISAT | WFM-DOAS (WFMD) | 4.0 | IUP, Univ. Bremen | Buchwitz et al., 2000; Schneising et al., 2011, 2012, 2013 |
| CH4_SCI_IMAP | SCIAMACHY on ENVISAT | IMAP-DOAS (IMAP) | 7.1 | JPL/SRON | Frankenberg et al., 2005, 2006, 2008a, 2008b, 2011 |
| CH4_GOS_OCPR | TANSO-FTS on GOSAT | UoL-Proxy | 6.0 | Univ. Leicester | Parker et al., 2011 |
| CH4_GOS_SRFP | TANSO-FTS on GOSAT | RemoTeC | 2.3.7 | SRON/KIT | Butz et al., 2011 |

**Table 1.** Overview of the used satellite $XCH_4$ data products.

| Source region | Latitude range [deg] | Longitude range [deg] | Mexp (*) [-] | Length L [km] | Overall conversion factor CF (*) [$MtCH_4$/yr/ppb] |
|---|---|---|---|---|---|
| Four Corners | 36.2 – 37.4 | 109.6W - 107.0W | 0.79 | 176.5 | 0.0518 |
| Central Valley (southern part) | 35.0 – 37.0 | 120.0W – 118.5W | 0.94 | 174.4 | 0.0605 |
| Azerbaijan | Country shape | | 0.94 | 294.3 | 0.1026 |
| Turmenistan | Country shape | | 0.98 | 698.6 | 0.2529 |

**Table 2.** Details related to the four source regions and their parameters as used for the emission estimation. (*) Approximate values (the exact values depend on the sampling of the satellite data in the source region, which depends on satellite product

10   and year).





| | Estimated methane emissions [MtCH$_4$/year] | | | | Comments / |
| Source region | SCIAMACHY | | GOSAT | | Other estimates |
| | WFMD | IMAP | OCPR | SRFP | |
|---|---|---|---|---|---|
| Four Corners | 0.50 [0.40, 0.59] | 0.57 [0.34, 0.80] | 0.45 [0.14, 0.76] | 0.42 [0.20, 0.64] | Kort et al., 2014 (*): 0.59 [0.54, 0.64] Turner et al., 2015: [0.45, 1.39] EDGAR v4.2: 0.17 |
| Central Valley (southern part) | 1.05 [0.53, 1.57] | 1.10 [0.92, 1.28] | 1.35 [0.96, 1.75] | 1.55 [1.15, 1.95] | EDGAR v4.2: 0.17 Jeong et al., 2013: 0.85 – 0.94 (for their region R12) |
| Azerbaijan | 0.60 [-0.01, 1.21] | 0.53 [0.23, 0.83] | 0.51 [-0.16, 1.18] | - | EDGAR v4.2 (FT2012): 0.74 |
| Turkmenistan | 1.89 [1.22, 2.55] | 1.93 [1.66, 2.19] | 2.08 [1.67, 2.49] | 1.85 [1.31, 2.39] | EDGAR v4.2 (FT2012): 1.33 |

**Table 3.** Summary of estimated methane emissions in terms of annual mean value and 1-sigma range obtained from computing the standard deviation of the annual emissions. (*) Kort et al., 2014, report the 2-sigma range [0.50, 0.67], not the (approximate) 1-sigma range listed here.




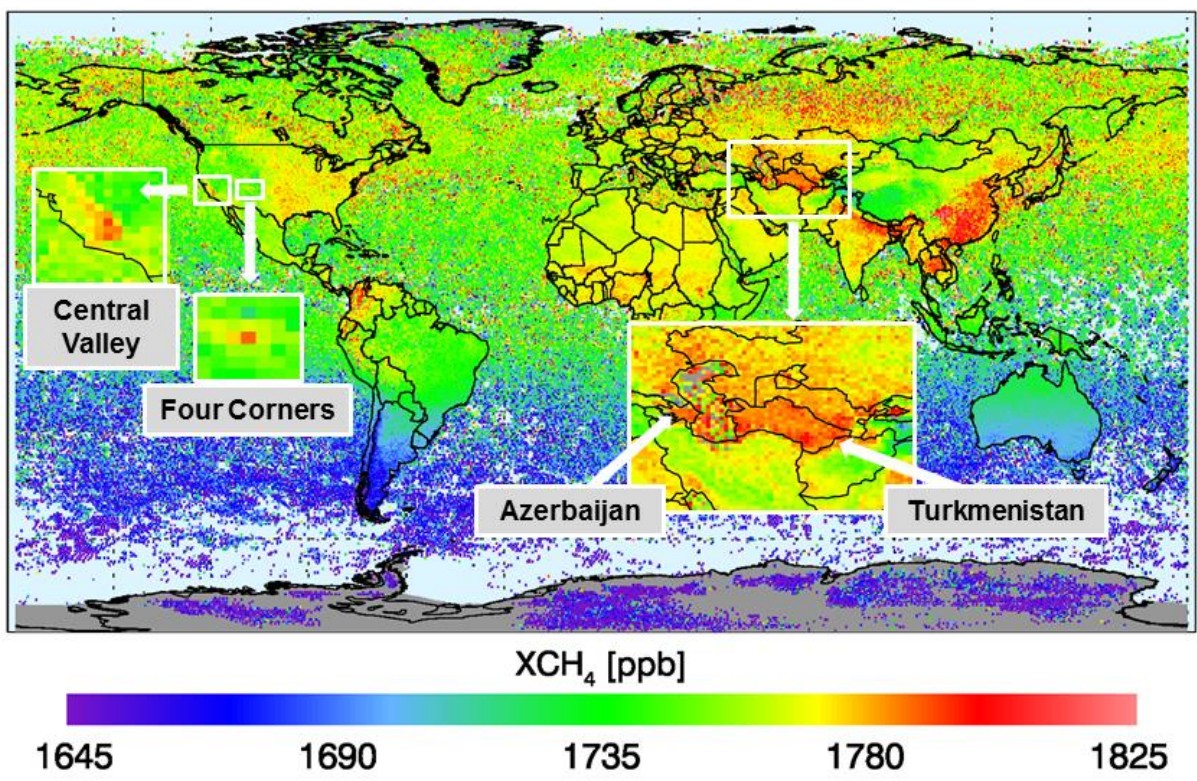

**Figure 1.** Year 2004 SCIAMACHY WFMD $XCH_4$ at $0.5^{\circ} \times 0.5^{\circ}$ resolution. The three target regions studied in this manuscript are indicated: Central Valley, California, USA, the Four Corners area in the southwestern USA, Azerbaijan and Turkmenistan.




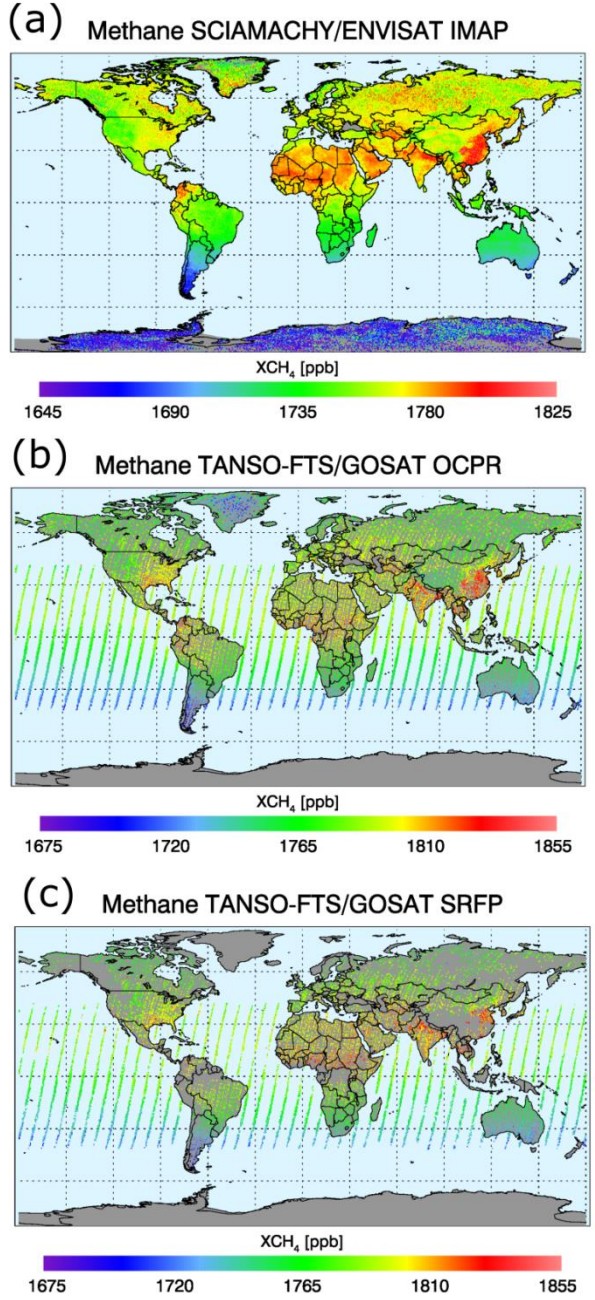

**Figure 2.** As Fig. 1 but for (a) SCIAMACHY IMAP XCH$_4$, (b) year 2010 GOSAT OCPR XCH$_4$ and (c) year 2010 GOSAT SRFP ("RemoTeC") XCH$_4$.




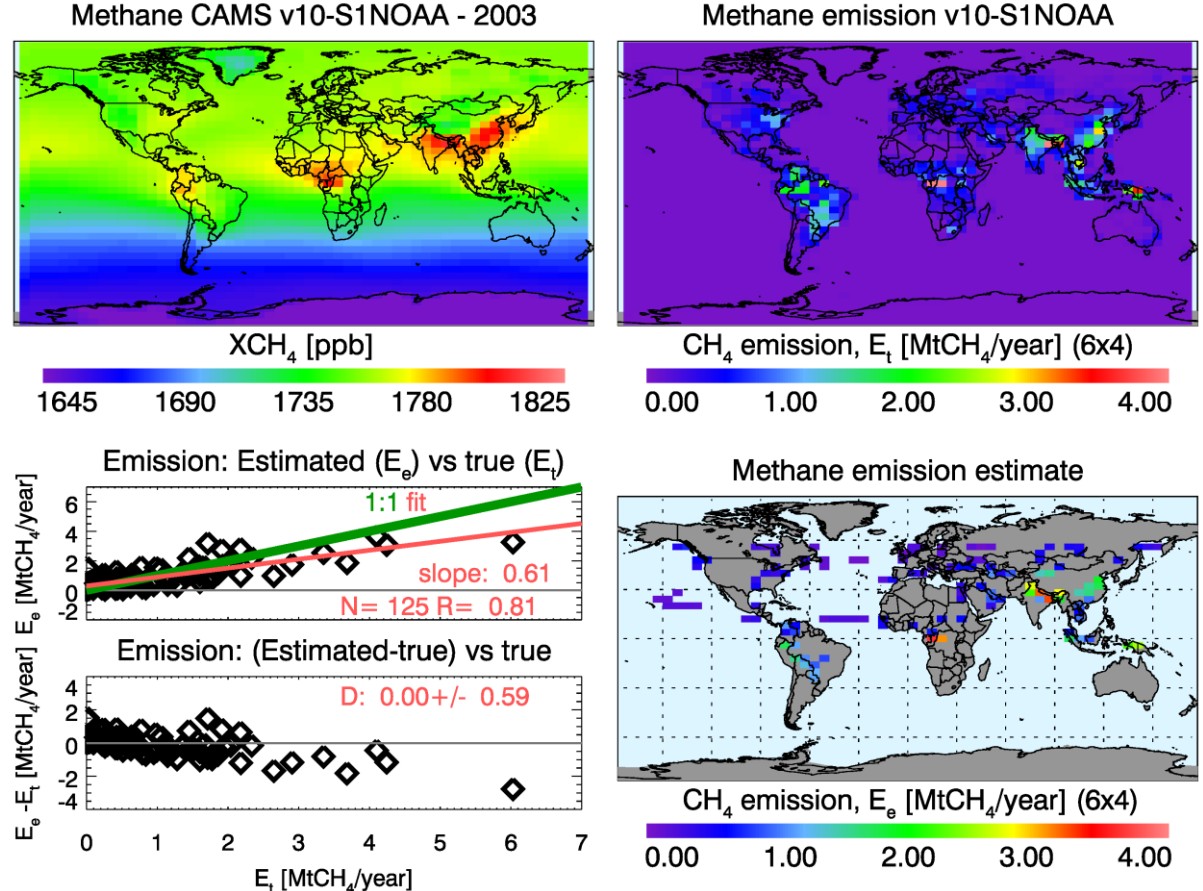

**Figure 3.** Methane emissions (in MtCH$_4$/year) and corresponding XCH$_4$ (in ppb) for the year 2003 at 6$^o$ longitude times 4$^o$ latitude resolution. Top left: XCH$_4$ as computed from Copernicus Atmosphere Monitoring Service (CAMS) atmospheric CH$_4$ fields (version v10-S1NOAA; resolution: 6$^o$x4$^o$; obtained from https://atmosphere.copernicus.eu/). Top right: Corresponding CAMS total, i.e., anthropogenic and natural, methane emissions. Map bottom right: Methane emissions of (automatically determined potential) emission hot spots ("hotspot cells") as derived from the top left XCH$_4$ map using the method described in Sect. 3. Bottom left: Comparison of retrieved emissions (map bottom right) with the "true" CAMS emissions (map top right). Here N (= 125) denotes the number of grid cells for which emission values have been obtained ("hotspot cells", see main text for details), R (= 0.81) is the linear correlation coefficient of retrieved and true emissions, and D is the difference between the retrieved and the true emissions in terms of mean difference and standard deviation (0.00 +/- 0.59 MtCH$_4$/year).



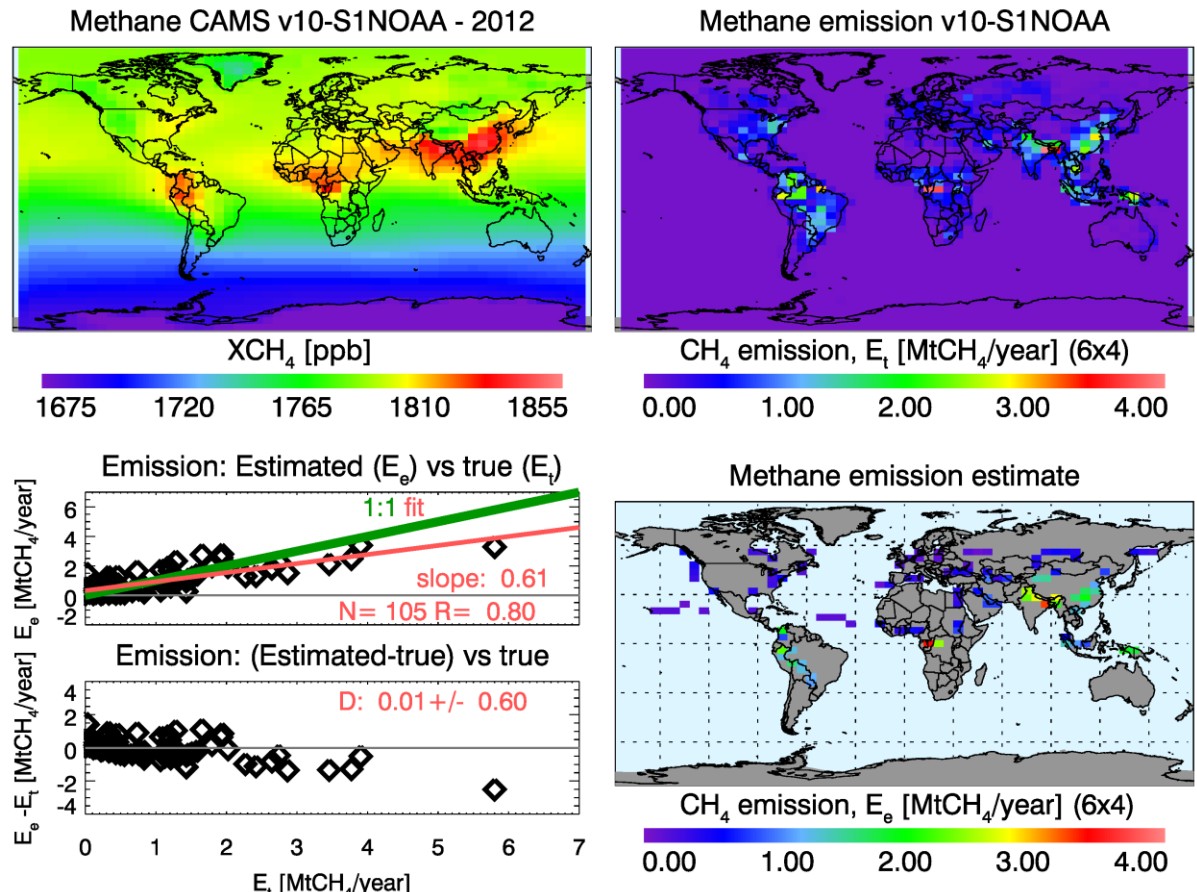

**Figure 4.** As Fig. 3 but for year 2012.



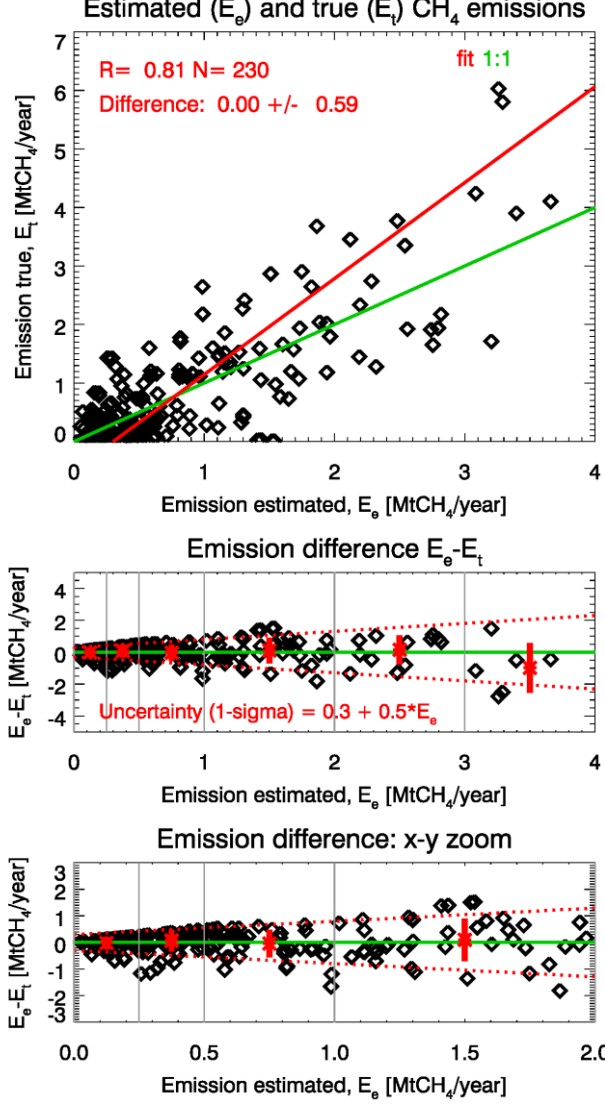

**Figure 5.** Top: "True" (i.e., CAMS) emission, $E_t$, versus estimated emissions, $E_e$, as obtained from the simulation-based assessment results shown in Figs. 3 and 4 (i.e., shown are all "hotspot cells" also shown in these two figures, see caption Fig. 3 and main text for details). Middle and bottom: Emission difference "estimated minus true" versus estimated emission. The grey vertical bars denote the boundaries of emission bins for which mean differences (red crosses) and standard deviations of the differences (red vertical lines) have been computed. The red dotted line shows that the relationship between the estimated emission ($E_e$) and its 1-sigma uncertainty ($\sigma$) can be approximately described by $\sigma(E_e) = 0.3 + 0.5\,E_e$.





**Figure 6.** Satellite-derived XCH$_4$ anomalies (i.e., the mean value of XCH$_4$ has been subtracted) in and around the Four Corners region. Top row: SCIAMACHY WFMD year 2004 XCH$_4$ anomaly at 0.5$^o$x0.5$^o$ resolution. (a) Originally gridded data. The black rectangle indicates the assumed source area (taken from Kort et al., 2014). (b) As (a) but after elevation correction (see main text for details). (c) As (b) but replacing the individual XCH$_4$ values by their averages in the indicated source region (inner rectangle) and its surrounding (outer rectangle). The difference between these two values defines the methane enhancement of the source region, i.e., $\Delta$XCH$_4$. Middle row: As top row but for IMAP XCH$_4$. Bottom row: As last column of first two rows but for GOSAT OCPR (g) and SRFP (h) for the year 2010.



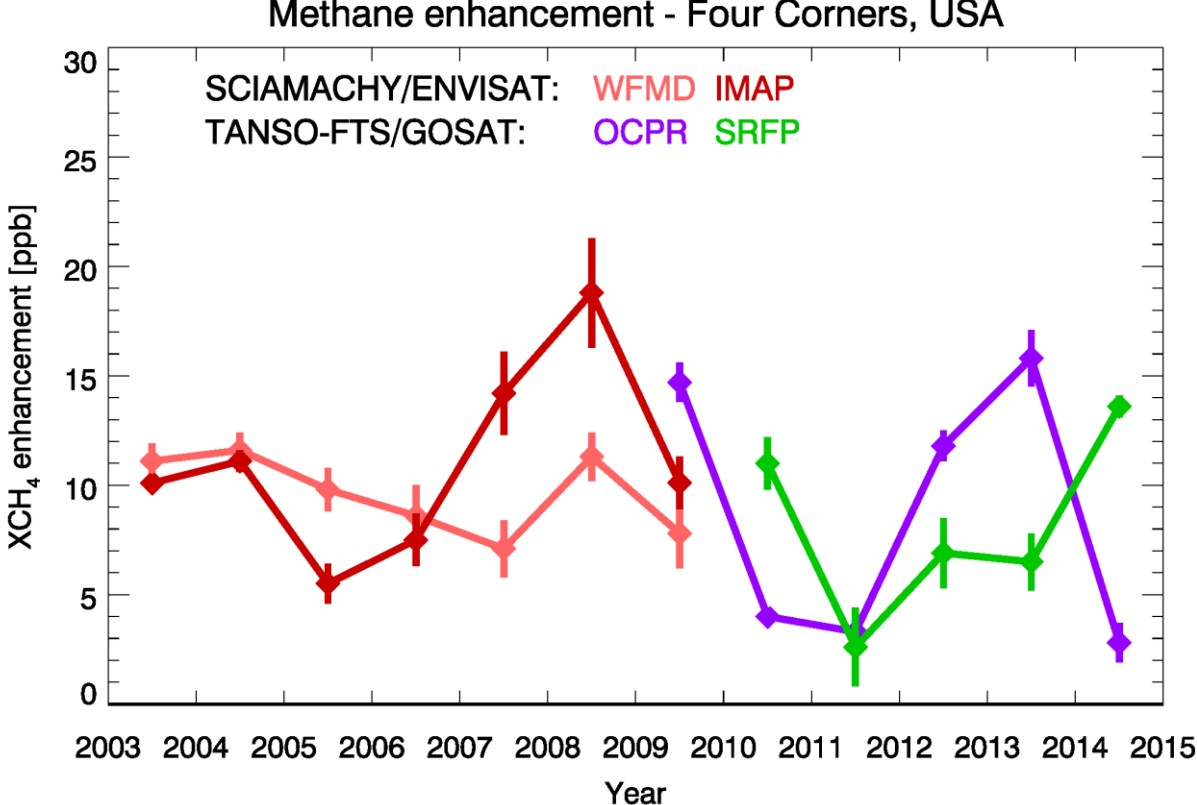

**Figure 7.** Methane enhancements over the Four Corners area for all years and all four satellite data products used in this study. The error bars show the standard deviation of the methane enhancements obtained by varying the size and shape of the surrounding area.



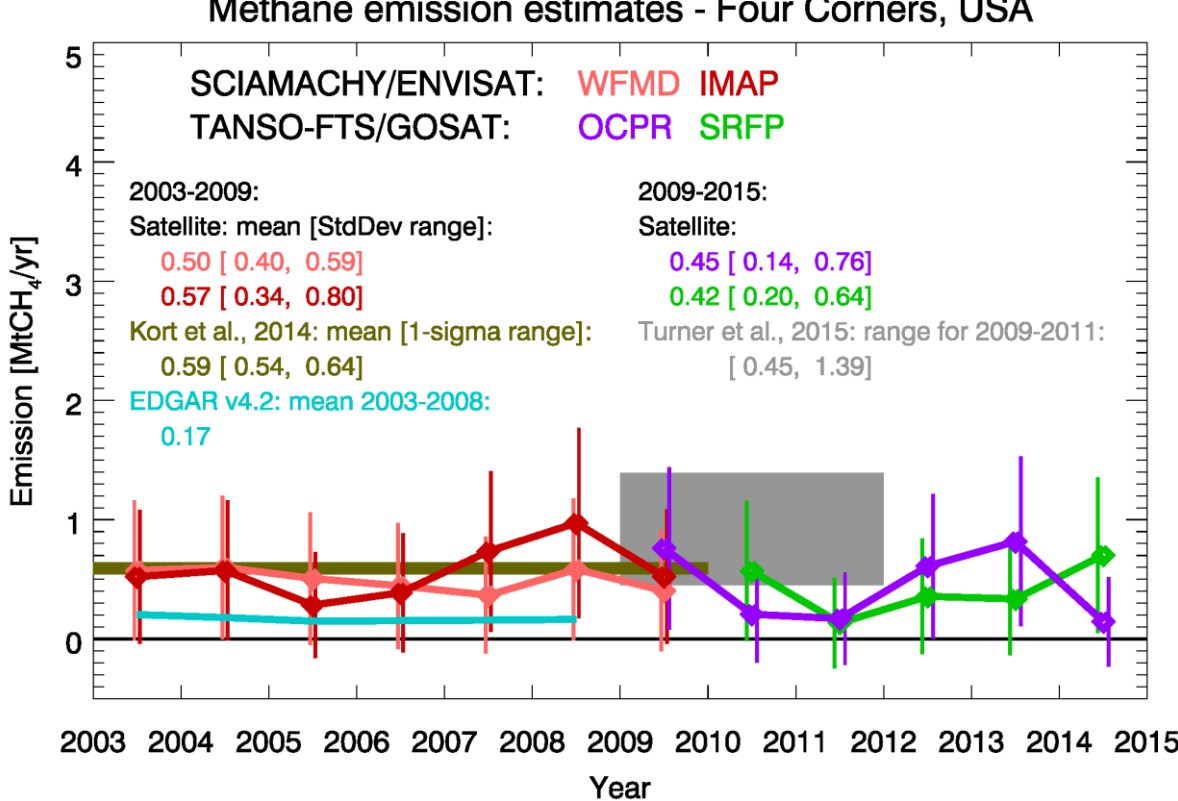

**Figure 8.** Methane emission estimates for Four Corners as obtained from the methane enhancements shown in Fig. 7. Shown
here are the satellite-derived annual methane emissions and their 1-sigma uncertainty as derived from the four satellite data
products used in this study using the method described in Sect. 3. The listed numerical values for the satellite-derived
emissions are the mean value and a range defined as mean value plus/minus one times the standard deviation of the annual
averages. The results are compared with published values as listed in Kort et al., 2014, (for 2003-2009; shown in dark green)
and Turner et al., 2015 (for 2009-2011; shown in grey). Also shown are the EDGAR v4.2 total anthropogenic emissions
during 2003-2008 (in light blue).



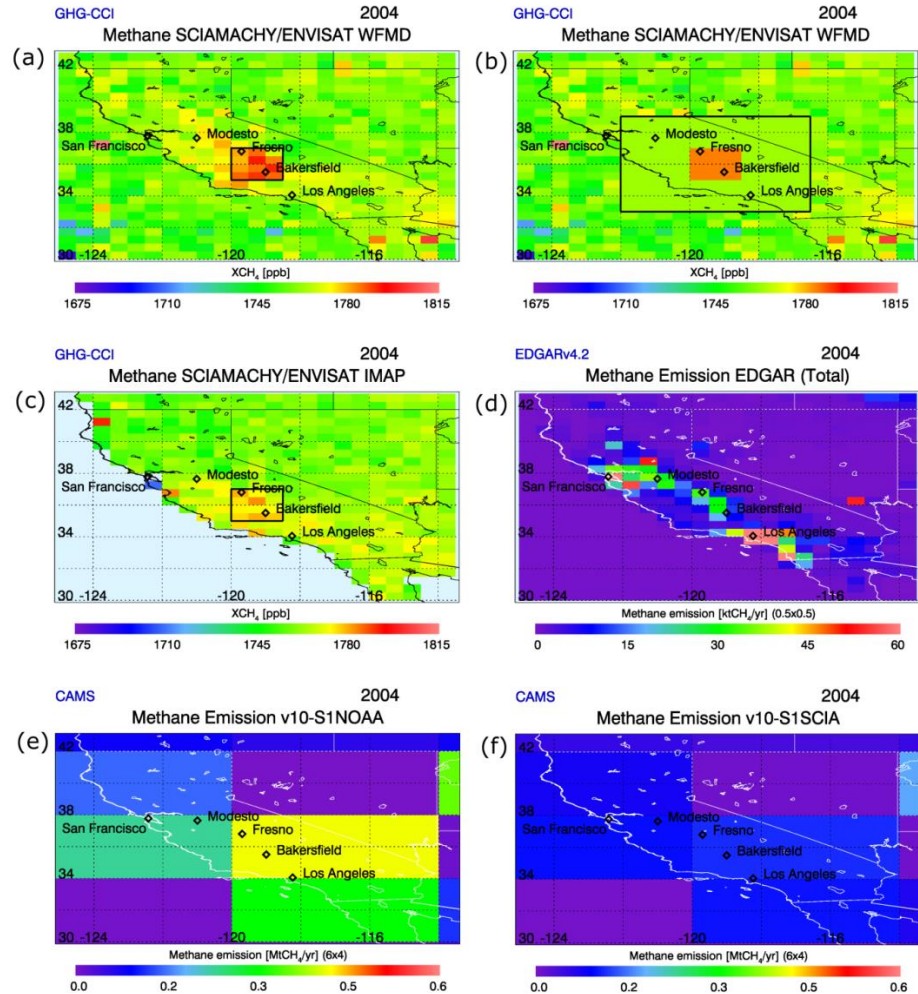

**Figure 9.** Methane maps for Central Valley, California. (a) SCIAMACHY year 2004 WFMD XCH$_4$ at 0.5$^o$x0.5$^o$ resolution. The rectangle shows the chosen source region. (b) As (a) but showing the source region (inner rectangle) and the default background region (outer rectangle) with their corresponding XCH$_4$ mean values. (c) As (a) but for IMAP. (d) EDGAR v4.2 year 2004 total anthropogenic methane emissions (regridded to 0.5$^o$x0.5$^o$ resolution). (e) CAMS v10-S1NOAA year 2004 total methane, i.e., anthropogenic and natural, emissions obtained by assimilation of NOAA methane observations (at 6$^o$x$^o$4). (f) As (e) but for CAMS version v10-S1SCIA, i.e., including the assimilation of SCIAMACHY IMAP retrievals in addition to the assimilation of NOAA data.



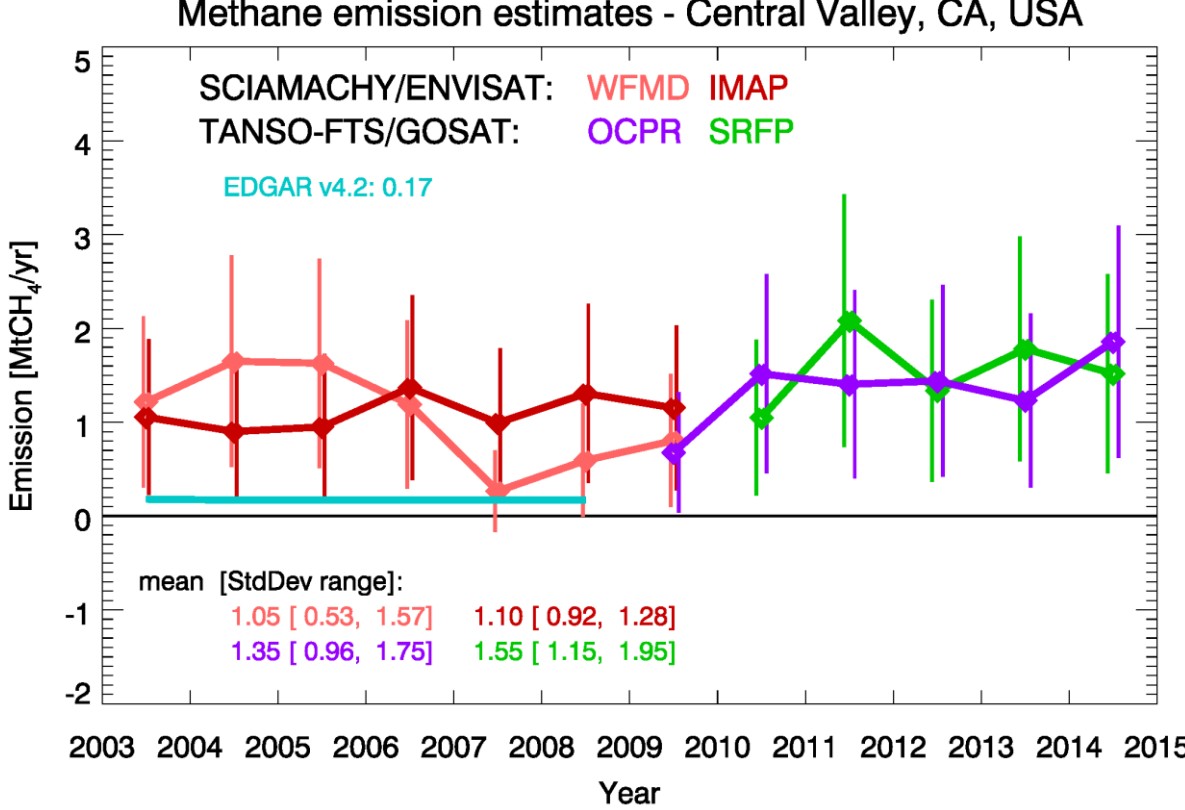

**Figure 10.** Methane emission estimates for Central Valley area in California, USA, as defined for this study (see Fig. 9 and Tab. 2). The blue line shows the EDGAR v4.2 (annual) anthropogenic methane emissions as computed for the Central Valley source region.





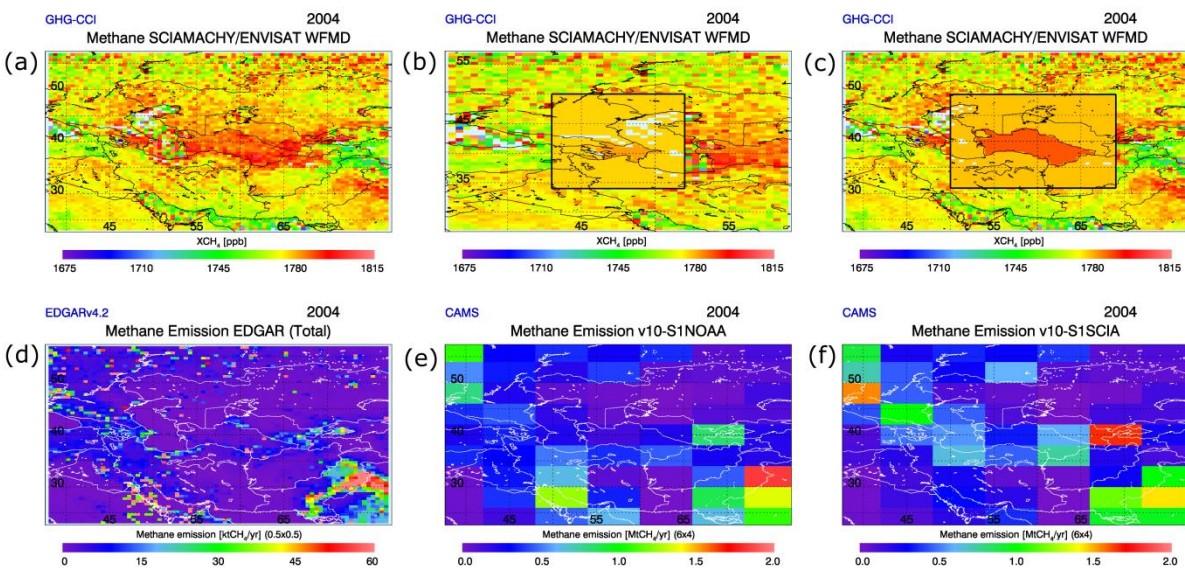

**Figure 11.** Top row: (a) SCIAMACHY WFMD year 2004 XCH$_4$ in and around Azerbaijan and Turkmenistan (resolution

5   0.5$^o$x0.5$^o$). (b) As (a) but showing the Azerbaijan source region (entire country of Azerbaijan) and the default background

region (rectangle) (note that this map is shifted relative to all other maps shown in this figure to place Azerbaijan in the

center of the map). (c) As (a) but showing the Turkmenistan source and default background regions. Bottom row: (d):

EDGAR v4.2 year 2004 total anthropogenic methane emissions (at 0.5$^o$x0.5$^o$ resolution). (e) CAMS year 2004 total

anthropogenic and natural methane emissions based on assimilation of NOAA data (at 6$^o$x4$^o$ resolution). (f) As (e) but with

10   additional assimilation of SCIAMACHY IMAP XCH$_4$.



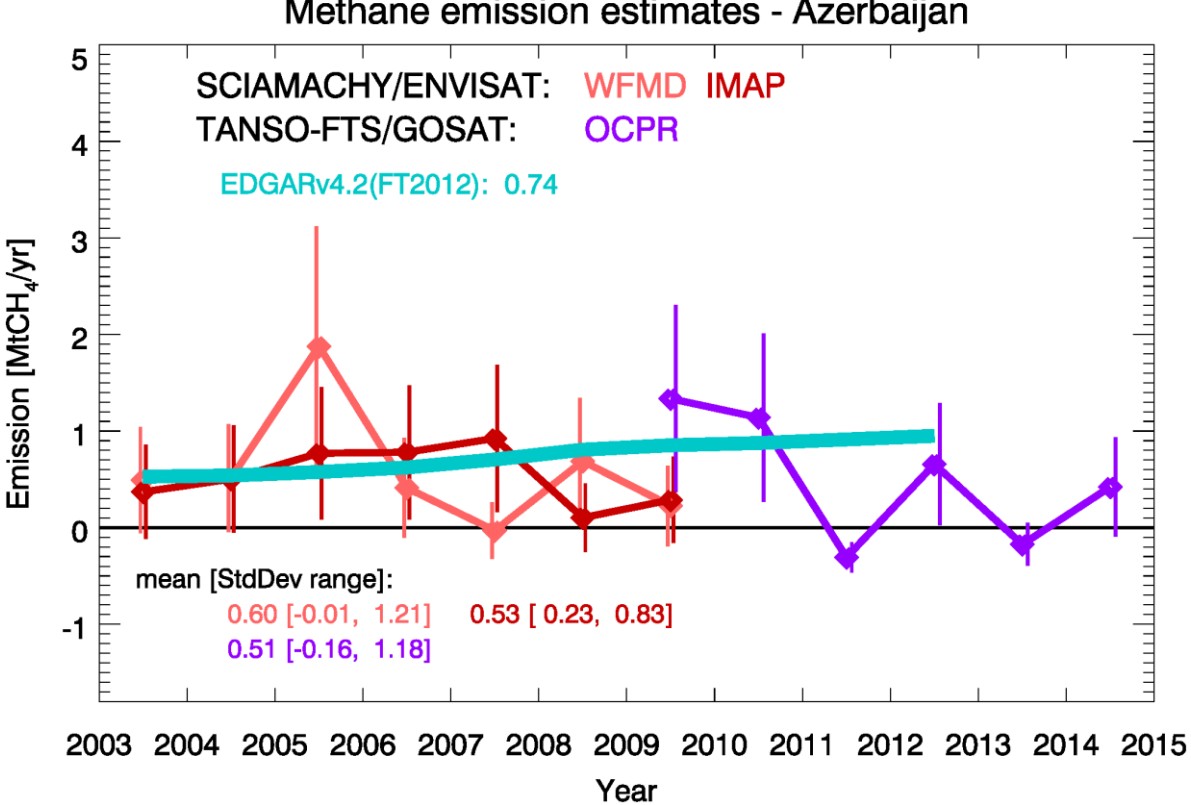

**Figure 12.** Methane emission estimates for Azerbaijan (see also Fig. 11). The blue line shows the EDGAR v4.2 (FT2012) emissions for Azerbaijan.





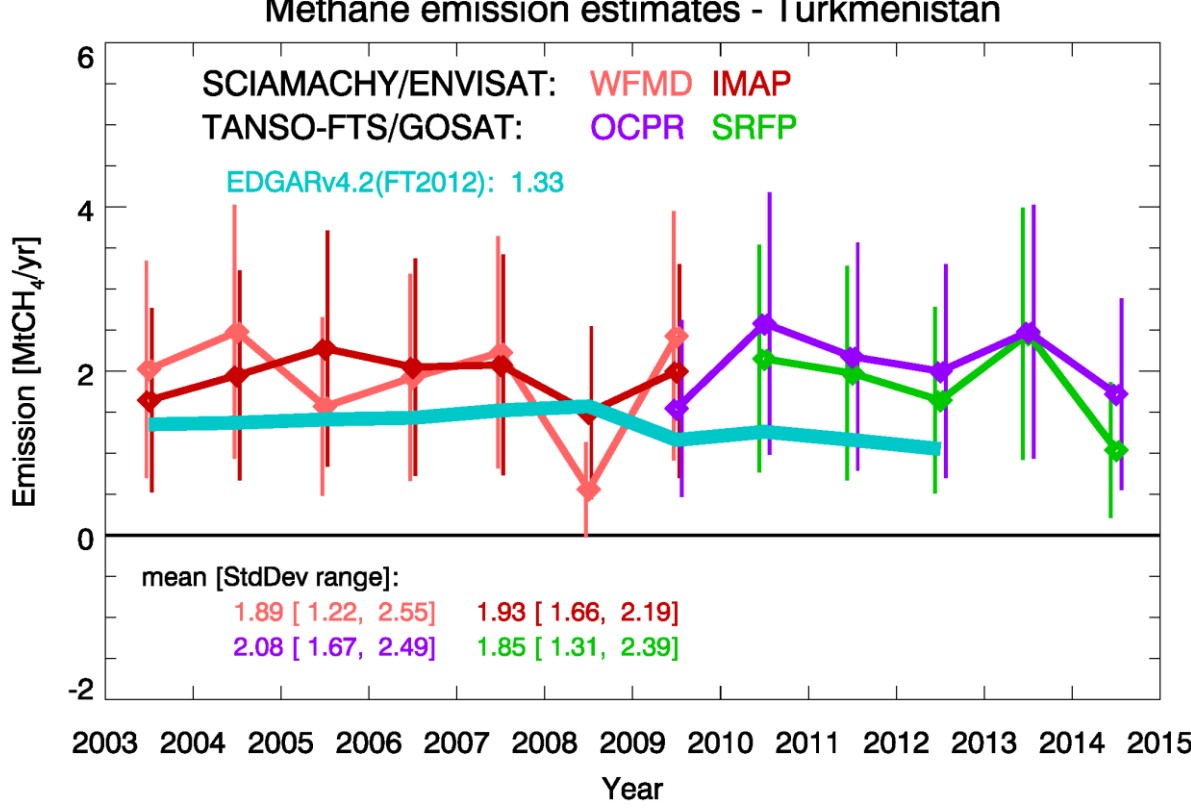

**Figure 13.** Methane emission estimates for Turkmenistan (see also Fig. 11). The blue line shows the EDGAR v4.2 (FT2012)

5   emissions for Turkmenistan.





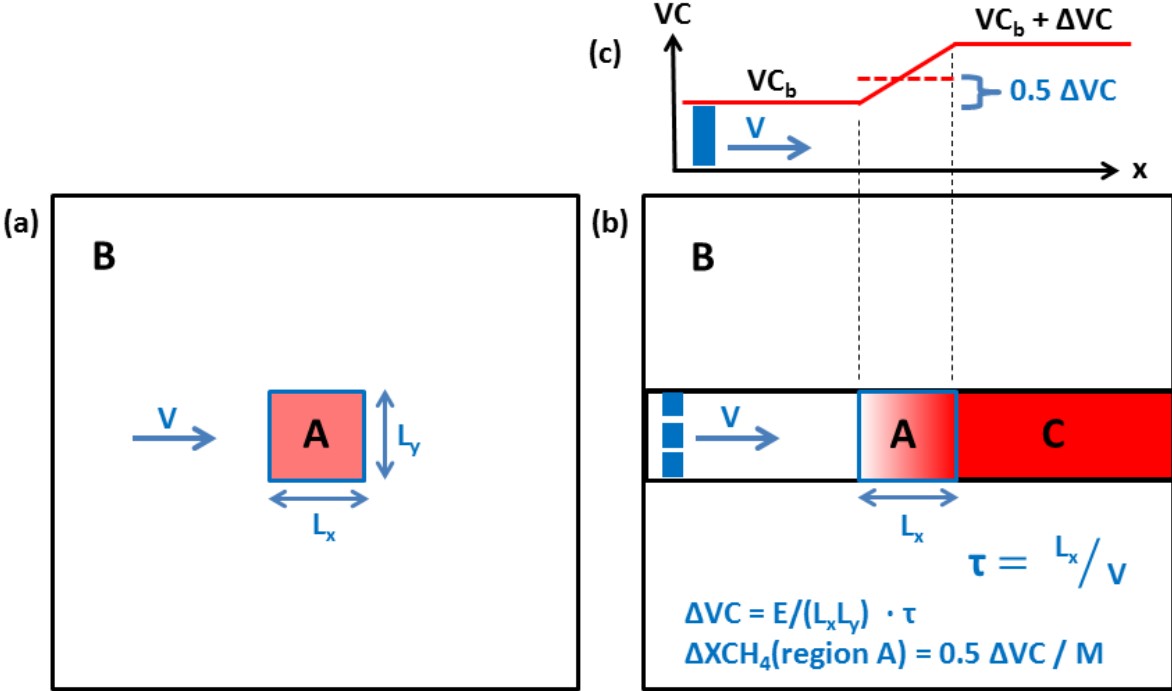

**Figure A1.** Sketch of a simple model used to explain the methane emission estimation method described in Sect. 3. (a) Source region A (of size $L_x L_y$ and with $L_x$ in wind speed direction (wind speed magnitude V)) with elevated $XCH_4$ (light red) and surrounding (background) region B. (b) Air parcels (blue squares) moving with constant speed V over a source region with emission $E/(L_x L_y)$, where E is the source area emission in $CH_4$ mass per time, while accumulating methane during accumulation time $\tau$ ($= L_x/V$). (c) Before entering the source region, the air parcels are characterized by a background methane vertical column, $VC_b$, in units of $CH_4$ mass per area. When leaving the source area their vertical column has been enhanced by $\Delta VC = E/(L_x L_y) \cdot \tau$. When passing over the source region, their vertical column increases linearly and, therefore, the average column enhancement over the source region is $0.5 \cdot \Delta VC$. VC ($CH_4$ mass per area) can be converted to $XCH_4$ (ppb) via a factor M (in units of mass and per area and per ppb).