# Peer review of "Satellite-derived methane hotspot emission estimates using a fast data-driven method"

_Atmospheric Chemistry and Physics, 2016_

## Referee Comment (RC1) · Anonymous Referee #1 · 26 Sep 2016

Overview: Buchwitz et al. present a manuscript describing a simple, fast method for estimating methane emissions from intensely emitting regions using annually averaged satellite observations of XCH4. They apply the method considering both SCIAMACHY and GOSAT methane data, and focus on four distinct regions, two in the US (Four Corners and CA central valley), as well as Turkmenistan and Azerbaijan. They compare results with the EDGAR inventory and other independent estimates, and discuss the possible utility of this 'hot-spot' tool with future space-based observations. The manuscript is appropriately place in ACP, writing is generally clear, the topic is interesting and the work is worthwhile. I do have some larger concerns with the work however, both in the appropriateness/robustness of the method (in particular the usage of one representative ventilation speed globally) and the representation of the findings (outlined below). Once the authors have satisfactorily addressed these concerns I would

reconsider the manuscript for publication in ACP.

Larger concerns: Methodological concerns: The authors need to explicitly state the necessary conditions for their approach to produce robust emissions estimates. What is the size of the region, size of xch4 signal, isolation from other sources, meteorological conditions, and emissions magnitude are necessary for the approach works?

The method the authors employ is essentially a very simple mass balance approach where the elevated methane levels are attributed to a necessary flux assuming a constant wind speed (ventilation time). (note supplemental figure A1 is actually very helpful in explaining the method and should really be in the main text). However, I was quite surprised that the author's determined one single wind speed for use around the globe in this technique. In essence, this states the size of the XCH4 enhancement seen in any hotspot is driven entirely by emissions, as wind speed is taken as globally constant. This would require significant justification, as we know this is not the case, and in particular, we know the manifestation of 'hot-spot' signal is often a consequence of meteorological conditions as well as emissions. For example, the Four Corners region discussed in the manuscript is known to exhibit pooling overnight, and part of what a midday satellite observations sees such an elevated signal is this meteorological dynamic (which is why in analyses such as the Kort et al., 2014 paper the winds are explicitly modeled). A region like North Dakota (discussed later), would have much higher wind speeds, and thus low XCH4 enhancements would actually be linked with higher emissions. There is much more justification needed to justify a single wind speed for all regions, as this would be expected to produce answers that are strongly biased at each individual region.

The comparison with the global model at 6x4 does not really provide a satisfactory answer as to why one wind speed would be appropriate – this analysis would suggest that integrating globally using one wind speed does not produce a biased estimated, but for individual regions (the whole point of the analysis) there can and will be large bias errors. Furthermore, calibrating with a model that is at 6x4 degrees would then

restrict the conclusion to analyses that are of the same resolution, as wind speeds in this type of box model setup will be rather different at a 6x4 degree region compared to a 1 or $\frac{1}{2}$ degree region.

How can you justify applying the analysis on such different spatial scales – small in CA and Four Corners and large areas in Turkmenistan and Azerbaijan? All of which are different scales than the 4x6 degree model used for calibration?

Why are these four regions chosen only? There should be some discussion of what selection bias may be present and the reasoning behind the choice.

Why have the author's ignored two other regions in the US which they have published on previously (Schneising et al., 2014 for North Dakota and Texas)? It is true the recent publication be Peischl et al., 2016 JGR collected aircraft data in North Dakota and showed the Schneising 2014 paper was physically inconsistent with the atmospheric observations and emissions estimates (and that it is implausible that emissions all of a sudden declined in the face of increasing production between the Schneising and Peischl studies)—but the authors here do not acknowledge that in citing the Schneising paper. One would suspect the discrepancy is because the Schneising paper relied on data from SCIAMACHY post-2009, which the author's have deemed not robust in this analysis. Given that this paper is discussing methane hotspots and an approach for quantifying a region and cites the Schneising paper, the North Dakota and Texas regions analyzed and published on previously by this group need to be addressed.

What emission model underlies the model runs used for simulation? Is that also EDGAR?

Representation problems: The abstract reads as if the paper provides a satellite estimate for emission in different regions that are statistically significantly different from best-estimate inventories for different regions (for example lines 26-27 about the central valley in CA). This is actually quite misleading. This oversells the utility and robustness of the conclusions compared to the rather heavily caveat-ed discussion in the main text.

Firstly, the authors imply through much of the text the uncertainty in their approach is often 100% or greater, and this is neglected in the abstract. Secondly, the central valley CA result is much larger than EDGAR, but is rather close to the best estimates made in the literature from both other top-down studies, but also from other bottom-up inventories specifically made for California! The authors cite and acknowledge this in the main text, but the abstract sensationalizes a 6-9x discrepancy with EDGAR, which is known to fail at these spatial scales and really does not mean reported or inventoried emissions are too low. In general, comparisons with EDGAR are fine to do, but should not be overemphasized as being thought of as an accurate representation of emissions on small spatial scales (or representative of government reported inventories on this scale).

Also, what is the overall utility of this method?

Where around the world can it be used?

Which regions satisfy the criterion for usage (and what is the criteria)?

What percentage of emissions can be tracked or observed this way? Need to see these numbers to understand the utility and impact of the approach.

Detailed comments: Page 1 line 26-30: these concluding sentences in the abstract are misleading and overstated as discussed above.

Page 2 line 24-26: This is where the question of selection bias and why these regions comes into play.

Page 7: This would be where defining the location requirements (ie XCH4 signal, size of area, wind speeds, emissions rate) would be valuable

Page 7 line 22: This is where the single wind speed is defined – see above for the concerns related to this approach.

Page 8 Line 8-9: This claim is really not robust. My assessment of these tests suggest

that integrating globally the single, constant wind speed does not lead to a (large) bias, but for individual regions it will be strongly biased and this must be addressed and fixed.

Page 10 Line 17-18: Agreement here does not indicate the approach is sound and robust and therefore can safely be applied. There could easily be errors that cancel and lead to a coincidental agreement, or it could be that this one region is particularly good for this method.

Page 10 Line 23-26: This type of comparison is misleading – the under representation of EDGAR on this small regions is well known and defined previously, and emphasizing this gives an inaccurate impression that these high emissions are not accounted for properly in inventories (on this spatial scale EDGAR does not agree or match even the US inventory).

Page 12 line 29: This is a prime example of why the assumed constant velocity globally is of concern. Four Corners experiences even more pooling of emissions than the Central Valley, yet that isn't discussed. This problem or wind speed representation gives great concern to this approach.

Page 12: Far to much discussion and emphasis on the comparison to EDGAR for the central valley. The emissions being higher there than in EDGAR is well understood and documented from top-down and bottom-up emissions estimates in citations referenced, and is more an illustration of the failure of EDGAR on small, sub-national scales.

Page 13 Line 21-22: If this is not a well-defined emission hotspot, why focus this study on this region?

Page 14 Line 28: typo, "toinvestigate"

Page 15 Line 7-9: This type of statement about concern about errors/problems in the approach needs to be addressed more explicitly in the abstract, and also should be addressed more quantitatively in sections such as this in the manuscript – what are the

possible magnitudes of bias errors?

---

## Referee Comment (RC2) · Anonymous Referee #2 · 17 Oct 2016

General comments:

This study uses satellite products for XCH4 to estimate regional methane emissions identifying hotspots. The authors tried to explain the method and results adding details, which I think help the reader understand the material. The topic is important and interesting, but I have a few major concerns about the method and the main result obtained from this proposed approach.

First, the word "hotspot" in the title is somewhat misleading because the main result of this study is a regional or subregional (relatively large area) estimate of CH4 emissions although pixels (but at coarse resolution) with large enhancements relative to surrounding pixels are identified. I strongly suggest that the authors remove the word "hotspot" from the title because this work essentially estimates emissions for source "regions",

for which many studies have already been doing using data from ground tower sites or aircrafts or remote sensing. The more accurate bottom-up inventories the authors cites (e.g., Jeong et al., 2014) can now identify hotspots with a much finer resolution. At the global scale, the source regions in this study may be considered hotspots, but those areas are really regions or subregions as shown in many previous studies.

Second, the authors try to match their satellite-based XCH4 to another assimilated product. This is disappointing because the value of those satellite products for XCH4 is significantly diminished as they are supposed to be used as independent retrievals of XCH4. The authors need a clear justification for this. Please see the related specific comments below.

Third, it looks like that the proposed method ends up with a simple linear scaling of satellite-derived XCH4 to CAMS, in particular with a single parameter of V, which seems to be estimated as one value for the whole globe (as written it sounds like that; if not please clarify it).

Also, given the too large uncertainty for individual annual emission estimates, I wonder what value from this study can be added to the scientific community for regional GHG modeling.

Specific comments:

Page 4, Lines 26 - 28, the sentence needs to be revised because the authors are trying to say two conflicting things in the sentence, making it confusing. Also, I would recommend that the authors be more quantitative instead of saying "agree reasonably well". In terms of data gap, how SCIAMACHY and GOSAT are different, e.g., available data points/pixels at the annual scale?

Page 5, Line 10, I wonder if the authors considered the data scarcity (i.e., small number of data) for the annual averages in terms of uncertainty. For certain pixels, the # of available data would be too small while others have enough for averaging.

Page 5, Line 17, I wouldn't use the term "enhancement" because the surrounding region is not equal to the CH4 "background" region, e.g., the Pacific region for the western US.

Page 6, Lines 4 - 9, Looking at Eq. (1), the authors are trying to estimate emissions (flux) for the source region using $\Delta$XCH4. But $\Delta$XCH4 is not exactly the local enhancement, but only the relative enhancement to the surrounding region, which itself has some local enhancements. This will lead to underestimation of the emissions for the source region.

Page 7, Lines 9 - 10, The authors confirm my point about the underestimation when using Eq. (1). The authors state that "we aim at quantifying the impact of the choice of the surrounding region by varying its size and shape." This makes it very hard to adopt the proposed method in other regions because it involves adjustments of size and shape, likely yielding multiple estimates and subsequently expanding the uncertainty.

Page 7, Lines 20 - 22, There are two important concerns about the method. First, I expected from the title that the satellite products would provide independent observations as in most of the top-down studies. It is not vey satisfactory to try to match estimates from another product, i.e., CAMS. Also, from what is written here, I find that a single value for V needs a serious justification. Also, I am not convinced why CAMS should provide "true" estimates. Can the CAMS estimates be truly representative of any of the study sites/regions? How well are they compared with the estimates from previous studies for those source regions (maybe the word "true" may not be appropriate here; otherwise needs clarification).

With respect to the optimization of V, this parameter optimization would be the key to this study. However, it seems that there is no explanation or consideration of the errors between the relationship between CAMS and XCH4, which can be defined as:

CAMS = f(XCH4, V) + err

where the function f is likely a linear one and err is the irreducible error (e.g., mean 0, normal error). Here for correct estimation of V, we need some independent estimates for err, similar to a linear regression case with errors.

Page 10, Lines 18 - 21, I differ with the authors. The too large uncertainty suggests that the method is not powerful. I would conclude that the only value of the satellite products used in this study is to provide auxiliary information derived from the column-averaged XCH4 which is linearly scaled to match another model product (rather than independent measurements).

Page 11, 33-34, Again, the uncertainty is too large. When we think about hotspots, we expect relatively unambiguous isolation of emissions. The papers cited in this work already estimated emissions for the region with much better uncertainty. What policy makers need is identification of hotspots at the level of km scales and emission estimates for those small regions to mitigate sources from them. However, in this study, even the regional annual total yields very large uncertainty. Is there any way to reduce the uncertainty, even at the annual scale?

Table 3. EDGAR v4.2 happens to estimate the same Mt CH4 for both Four Corners and the Central Valley?

Figure 1. The region needs to be defined more accurately. For example, the region defined as the Central Valley of California in Figure 1 includes Southern California, and is different from that in Table 2.

Figure 8 needs some improvements. First, the data points (circles) should match the years on the X-axis label that are represented. Is the "standard deviation" the standard deviation of 7 annual estimates, e.g., for the 2003 - 2009. If this is the case, standard deviation is not very useful. I would be more interested in knowing the overall mean estimate for the multi-year period and the uncertainty about the mean, e.g., during 2003 - 2009. When individual annual estimates have huge uncertainties associated, I don't see the benefit of using standard deviation.

Also, the 1-sigma uncertainty in estimated emissions for individual years overlap with the EDGAR estimate, making it hard to statistically evaluate EDGAR. Looking at this at face value, I am not sure if there is any statistical power in the proposed method to say about the regional emission, even at the annual scale.

---

## Author Response (AR1)

Michael Buchwitz on behalf of all co-authors                    8-February-2017

**Authors response to referee comments on manuscript "Satellite-derived methane hotspot emission estimates using a fast data-driven method" of Michael Buchwitz et al., MS No.: acp-2016-755**

This document includes our point-by-point response to the reviews, a list of all relevant changes made in the manuscript, and a marked-up manuscript version.

**Point-by-point response:**

Our point-by-point response to the reviews has been submitted via the ACP website and is already online, see http://www.atmos-chem-phys-discuss.net/acp-2016-755/#discussion :

AC1: 'Authors response to comments from reviewer No. 1', Michael Buchwitz, 08 Feb 2017:

http://editor.copernicus.org/index.php/acp-2016-755-AC1.pdf?_mdl=msover_md&_jrl=10&_lcm=oc108lcm109w&_acm=get_comm_file&_ms=54349&c=118994&salt=554266951424103570

AC2: 'Authors response to comments from reviewer No. 2', Michael Buchwitz, 08 Feb 2017:

http://editor.copernicus.org/index.php/acp-2016-755-AC2.pdf?_mdl=msover_md&_jrl=10&_lcm=oc108lcm109w&_acm=get_comm_file&_ms=54349&c=118995&salt=621007627688187377

Nevertheless, these 2 documents with our answers are attached to this document (see following pages).

**Marked-up manuscript version:**

The Marked-up manuscript version is also attached to this document (at the end).

**List of all relevant changes:**

We have aimed at carefully addressing all referee comments (see our Point-by-point response). This resulted in several major and minor modifications which have been implemented for the revised version of our manuscript (see Marked-up manuscript version). The most relevant changes are:

- We have used an additional methane data set to address several of comments of the referees. This new high-resolution methane data set has been provided by Alexander J. Turner from the Harvard University, who has been added as a co-author. The new data set and its analysis is now described in a new Section 3.1, which also contain several (new) figures (Figs. 8-13).
- Based on the recommendation from one of the referees we have removed the Annex and moved the corresponding text and the figure to Sect. 3.

- Based on the comments from both referees we have added more information on our analysis w.r.t. the potential use of mean wind speed from meteorological data to improve the accuracy of our inversion method. This resulted in additional text in Sect. 3 and a new figure 7.
- Furthermore, there a many other changes: For example, we have improved text and figures at various places to eliminate typos and to improve explanations (as requested by the referees). We have also added an additional co-author from Univ. Leicester who is one of the data provider and helped to significantly improve the manuscript.

We conclude that addressing the referee comments resulted in a significantly improved version of our manuscript. We hope that the revised version of the manuscript meets the high standards of ACP.

Michael Buchwitz on behalf of all co-authors

One the following pages please see our response to the two referees and the marked-up manuscript version.

Michael Buchwitz, 8-Feb-2017

**Reply to reviewer No 1**

We thank the referee for carefully reading our manuscript and for providing a critical review. Below we are giving our point-by-point answers to each of the referee's comments and concerns. Addressing these comments, concerns and questions helped us to prepare a significantly improved version of our manuscript.

Q1: Referee:
Larger concerns: Methodological concerns: The authors need to explicitly state the necessary conditions for their approach to produce robust emissions estimates. What is the size of the region, size of xch4 signal, isolation from other sources, meteorological conditions, and emissions magnitude are necessary for the approach works?

Author's reply:
In the revised version of our manuscript we present additional investigations concerning the performance of our method. These investigations are based on a simulated high-resolution methane data set. Furthermore, we now better highlight already in the abstract limitations of our method. We show that our method typically provides a conservative estimate of the emissions, i.e., our emissions are typically underestimated. We better explain why our method tends to underestimate emissions. The large uncertainty of our method is reflected in quite large uncertainty estimates which are typically on the order of 100% for the source regions discussed in our manuscript. Our fast and simple method has been developed to obtain a reasonable estimate of the annual methane emission of a region which shows elevated methane relative to its surrounding region in maps of annually averaged satellite-derived $XCH_4$. When applying our method to multiple years of satellite data, the results will show, if elevated atmospheric methane is present in all years or not. If the methane is elevated in all years than this is very likely due to an "underlying" methane emission source (assuming that the satellite data do not have a "local bias"). We would not apply our method to situations, where this is not the case (although our method can be applied also to methane fields which are spatially constant/flat but in this case our method will deliver an estimated emission of zero together with a large error bar). There are no limitations w.r.t. the size of the region (as size is (approximately) considered by parameter L) or the size of the $XCH_4$ signal (as explained above) or the magnitude of the emission (which is unknown as the satellite only provides $XCH_4$). As shown in our manuscript we have not identified any conditions, where the method is shown to fail entirely but we recommend to be careful if the targeted source region is known to exhibit "pooling overnight" (more details on this aspect are given below) and/or for regions with complex topography (where, for example, methane can accumulate in valleys; these are however situations where all inversion methods will likely have severe difficulty). Our method assumes that the emission sources are homogeneously distributed in the targeted source region. We show in our manuscript that the estimated emissions are underestimated if this is not the case. Underestimation also results from sources located in the surrounding region. From all this we conclude that our method can be applied to all situations but that typically the emission will be underestimated, i.e., our estimates are conservative estimates. If the resulting emission is unexpectedly high, then this is a strong indication that the true emission is in fact higher than expected. In this case we recommend additional investigations, e.g., using a much more advanced (and computationally much more expensive) model than our simple mass balance approach (as explained in our paper).

Q2: Referee:
The method the authors employ is essentially a very simple mass balance approach where the elevated methane levels are attributed to a necessary flux assuming a constant wind speed (ventilation time). (note supplemental figure A1 is actually very helpful in explaining the method and should really be in the main text). However, I was quite surprised that the author's determined one single wind speed for use around the globe in this technique. In essence, this states the size of the XCH4 enhancement seen in any hotspot is driven entirely by emissions, as wind speed is taken as globally constant. This would require significant justification, as we know this is not the case, and in particular, we know the manifestation of 'hot-spot' signal is often a consequence of meteorological conditions as well as emissions. For example, the Four Corners region discussed in the manuscript is known to exhibit pooling overnight, and part of what a midday satellite observations sees such an elevated signal is this meteorological dynamic (which is why in analyses such as the Kort et al., 2014 paper the winds are explicitly modeled). A region like North Dakota (discussed later), would have much higher wind speeds, and thus low XCH4 enhancements would actually be linked with higher emissions. There is much more justification needed to justify a single wind speed for all regions, as this would be expected to produce answers that are strongly biased at each individual region.

Author's reply:

We are not assuming a constant wind speed. Wind speed is a parameter of our inversion model. However, we show that the consideration of (regionally varying) annual mean wind speed (as obtained from meteorological data) does not help to reduce systematic errors of our annual emissions as obtained from annually averaged $XCH_4$ (which is the goal of our method). We therefore use a constant wind speed but this is not because we assume this but because this results from our analysis, which shows that the use of spatially resolved annual mean wind speed (from meteorological data) does not help to improve our method. In the revised paper we present more details on this.

We removed Appendix A and present figure A1 now directly in our methods section.

"Pooling overnight" is in fact a concern for our method as this could result in a significant overestimation of the estimated emissions, which is what we aim to avoid as this would result in "false alarm" in comparison to emission inventories. For Four Corners we have no indication for overestimation of the Four Corners emissions as estimated with our method. In the revised version of our manuscript we investigated this using high-resolution methane simulations and found underestimation in line with the general characteristics of our method, which tends to underestimate emissions. We also present new results for several other regions (incl. California) and never found significant overestimation.

Q3: Referee:
The comparison with the global model at 6x4 does not really provide a satisfactory answer as to why one wind speed would be appropriate – this analysis would suggest that integrating globally using one wind speed does not produce a biased estimated, but for individual regions (the whole point of the analysis) there can and will be large bias errors. Furthermore, calibrating with a model that is at 6x4 degrees would then restrict the conclusion to analyses that are of the same resolution, as wind speeds in this type of box model setup will be rather different at a 6x4 degree region compared to a 1 or 1/2 degree region.

Author's reply:
In the revised version of our manuscript we address this aspect by presenting additional results based on high resolution (< 1 deg) methane simulations.

Q4: Referee:
How can you justify applying the analysis on such different spatial scales – small in CA and Four Corners and large areas in Turkmenistan and Azerbaijan? All of which are different scales than the 4x6 degree model used for calibration?

Author's reply:
In the revised version of our manuscript we address this aspect by presenting additional results based on high resolution (< 1 deg) methane simulations applied to small regions such as Four Corners and large (country-scale) regions such as large parts of California.

Q5: Referee:
Why are these four regions chosen only? There should be some discussion of what selection bias may be present and the reasoning behind the choice.

Author's reply:
We selected these four regions because they show up as regions of elevated methane in the satellite data products (e.g., our Fig. 1) and because they are extensively discussed in the peer-reviewed literature (Central Valley, CA, and Four Corners) or other data sets exist which can be used for comparison (e.g., EDGAR for Turkmenistan and Azerbaijan). Initially our main motivation to develop our method was to at least roughly estimate Turkmenistan's emissions as this country prominently shows up as a region of elevated methane in the satellite data. We also studied Azerbaijan because it is located close to Turkmenistan primarily to see how the estimated emissions of these two counties compare (as typically relative accuracy is better than absolute accuracy).

Q6: Referee:
Why have the author's ignored two other regions in the US which they have published on previously (Schneising et al., 2014 for North Dakota and Texas)? It is true the recent publication be Peischl et al., 2016 JGR collected aircraft data in North Dakota and showed the Schneising 2014 paper was physically inconsistent with the atmospheric observations and emissions estimates (and that it is implausible that emissions all of a sudden declined in the face of increasing production between the Schneising and Peischl studies) but the authors here do not acknowledge that in citing the Schneising paper. One would suspect the discrepancy is because the Schneising paper relied on data from SCIAMACHY post-2009, which the author's have deemed not robust in this analysis.

Given that this paper is discussing methane hotspots and an approach for quantifying a region and cites the Schneising paper, the North Dakota and Texas regions analyzed and published on previously by this group need to be addressed.

Author's reply:
As explained in our manuscript our method has been developed to obtain emission estimates for regions where satellite $XCH_4$ is clearly elevated compared to their surrounding areas. This condition is not met for the areas studied in Schneising et al., 2014 (see their Fig. 3). Furthermore, Schneising et al., 2014, used a method to minimize the potential impact of systematic errors of the used satellite product in later years by analysing differences of the satellite product between two 3-year time periods including years we are not analysing in our manuscript for reasons explained in our manuscript. We have therefore not ignored the two areas studied in Schneising et al., 2014, but we do not study them here because our method is not optimized to deal with them, in contrast to the method of Schneising et al., 2014.

We may misunderstand you but it appears that your comment suggests that it is "true" that the Schneising et al., 2014, results are "physically inconsistent" with other published observations. We do not agree with this as these other observations have not been made during the time period analysed by Schneising et al., 2014, but later and because such a statement needs to consider the uncertainty estimates as reported in Schneising et al., 2014. The uncertainty estimates as reported in Schneising et al., 2014, are large (nearly 70% 1-sigma) and statements w.r.t. consistency or inconsistency need to consider this. If one would do that one would find out that there is no inconsistency at a 5% (or even much higher) significance level. Concerning "that it is implausible that emissions all of a sudden declined" please see Schwietzke et al., Nature, 2016, showing that leakage rates of fugitive emissions decline with time.

Reference:
Schwietzke et al., Upward revision of global fossil fuel methane emissions based on isotope database, Nature, Vol. 539, 88-91, doi:10.1038/nature19797, 2016.

Q7: Referee:
What emission model underlies the model runs used for simulation? Is that also EDGAR?

Author's reply:
Several emission data bases are used as input as explained in Bergamaschi et al., 2009, but the anthropogenic a priori emissions are based on EDGAR. These emissions are however not used directly for the data set we used as this data set is based on forward modelling using optimize (a posteriori) emissions.

Q8: Referee:
Representation problems: The abstract reads as if the paper provides a satellite estimate for emission in different regions that are statistically significantly different from best-estimate inventories for different regions (for example lines 26-27 about the central valley in CA). This is actually quite misleading. This oversells the utility and robustness of the conclusions compared to the rather heavily caveat-ed discussion in the main text.

Author's reply:
For the revised version of our manuscript we have modified the abstract to also highlight the limitations of our method. We tried to eliminated all potential misunderstandings and clearly do not want to oversell our method and results.

Q9: Referee:
Firstly, the authors imply through much of the text the uncertainty in their approach is often 100% or greater, and this is neglected in the abstract. Secondly, the central valley CA result is much larger than EDGAR, but is rather close to the best estimates made in the literature from both other top-down studies, but also from other bottom-up inventories specifically made for California! The authors cite and acknowledge this in the main text, but the abstract sensationalizes a 6-9x discrepancy with EDGAR, which is known to fail at these spatial scales and really does not mean reported or inventoried emissions are too low. In general, comparisons with EDGAR are fine to do, but should not be overemphasized as being thought of as an accurate representation of emissions on small spatial scales (or representative of government reported inventories on this scale).

Author's reply:
For the revised version of our manuscript we have modified the abstract to also highlight the limitations of our method, e.g., by explicitly stating that uncertainty is often around 100%.

Also, what is the overall utility of this method?

Author's reply:
The overall utility of our method lies in the fact that it provides at least rough estimates of emissions of source regions from large amounts of satellite data. As explained in our paper, we recommend further studies using more complex (and therefore computationally much more expensive) methods in case our method indicates significantly higher emissions compared to emission inventories. We write in our "Summary and conclusions" section: "More detailed assessments likely require the use of much more complex approaches compared to the simple method used in this study. Nevertheless, simple and fast approaches also have a role to play as they permit to perform quick assessments on possible discrepancies with respect to emission inventories or other data sets and can also be used for plausibility checks for more complex approaches".

Q11: Referee:
Where around the world can it be used?

Author's reply:
We have not identified any region where it cannot be used but please see also our detailed response to your first concern Q1.

Q12: Referee:
Which regions satisfy the criterion for usage (and what is the criteria)?

Author's reply:
We have not identified any region where our method cannot be used but please see also our response to your earlier questions but in particular our detailed response to your first concern Q1.

Q13: Referee:
What percentage of emissions can be tracked or observed this way? Need to see these numbers to understand the utility and impact of the approach.

Author's reply:
This question is difficult to answer but in general (as shown in more detail in the revised version of our manuscript) the local or regional emission sources must be quite strong, on the order of several 100 $ktCH_4$/yr. As also shown in the revised version of our manuscript there are many of these source regions in the USA and therefore likely also in many other parts of the world. However, we cannot give a reliable number in terms of percent of total methane emissions at this stage as this answer also depends on the spatial distribution of the sources (as our method requires relatively well isolated sources).

Q14: Referee:
Page 1 line 26-30: these concluding sentences in the abstract are misleading and overstated as discussed above.

Author's reply:
We have modified the abstract to highlight also limitations of our method and we explicitly mention that our uncertainty is on the order of 100%.

Q15: Referee:
Page 2 line 24-26: This is where the question of selection bias and why these regions comes into play.

Author's reply:
Please see our answers as given above on these aspects.

Q16: Referee:
Page 7: This would be where defining the location requirements (ie XCH4 signal, size of area, wind speeds, emissions rate) would be valuable

Author's reply:
Please see our answers as given above on these aspects.

Q17: Referee:
Page 7 line 22: This is where the single wind speed is defined – see above for the concerns related to this approach.

Author's reply:
Please see our answers as given above on these aspects.

Q18: Referee:
Page 8 Line 8-9: This claim is really not robust. My assessment of these tests suggest that integrating globally the single, constant wind speed does not lead to a (large) bias, but for individual regions it will be strongly biased and this must be addressed and fixed.

Author's reply:
Please see our answers as given above on these aspects and please note that in the revised version of the manuscript we present additional investigations using high-resolution methane simulations and we apply our method to these simulations to obtain a better understanding of the performance of our method.

Q19: Referee:
Page 10 Line 17-18: Agreement here does not indicate the approach is sound and robust and therefore can safely be applied. There could easily be errors that cancel and lead to a coincidental agreement, or it could be that this one region is particularly good for this method.

Author's reply:
Yes, it is true that this could be a coincidental agreement. Therefore, we added for the revised version of the manuscript additional investigations using high-resolution methane simulations and we also added Four Corners to this extended assessment.

Q20: Referee:
Page 10 Line 23-26: This type of comparison is misleading – the under representation of EDGAR on this small regions is well known and defined previously, and emphasizing this gives an inaccurate impression that these high emissions are not accounted for properly in inventories (on this spatial scale EDGAR does not agree or match even the US inventory).

Author's reply:
EDGAR is an important, frequently used and carefully constructed data base (which does not mean that EDGAR is perfect) and we have not found statements in the peer-reviewed literature that EDGAR is inaccurate and therefore should not be used for applications like this.

Q21: Referee:
Page 12 line 29: This is a prime example of why the assumed constant velocity globally is of concern. Four Corners experiences even more pooling of emissions than the Central Valley, yet that isn't discussed. This problem or wind speed representation gives great concern to this approach.

Author's reply:
Please see our response as given above.

Q22: Referee:
Page 12: Far to much discussion and emphasis on the comparison to EDGAR for the central valley. The emissions being higher there than in EDGAR is well understood and documented from top-down and bottom-up emissions estimates in citations referenced, and is more an illustration of the failure of EDGAR on small, sub-national scales.

Author's reply:
Please see our response w.r.t. EDGAR as given above.

Q23: Referee:
Page 13 Line 21-22: If this is not a well-defined emission hotspot, why focus this study on this region?

Author's reply:
Please see our response w.r.t. Turkmenistan as given above.

Author's reply:
Many thanks. This has been corrected.

Q25: Referee:
Page 15 Line 7-9: This type of statement about concern about errors/problems in the approach needs to be addressed more explicitly in the abstract, and also should be addressed more quantitatively in sections such as this in the manuscript – what are the possible magnitudes of bias errors?

Author's reply:
For the revised version this comment has been considered by modifying the abstract and by providing additional investigations using high-resolution methane simulations and more detailed discussion at several places.

Michael Buchwitz, 8-Feb-2017

**Reply to reviewer No 2**

First of all, we would like to thank the referee for carefully studying our manuscript and for providing critical comments and questions. Below we provide answers to each of these comments and questions. The referee's comments have resulted in a significantly improved version of our manuscript.

Q1: Referee:
First, the word "hotspot" in the title is somewhat misleading because the main result of this study is a regional or subregional (relatively large area) estimate of CH4 emissions although pixels (but at coarse resolution) with large enhancements relative to surrounding pixels are identified. I strongly suggest that the authors remove the word "hotspot" from the title because this work essentially estimates emissions for source "regions", for which many studies have already been doing using data from ground tower sites or aircrafts or remote sensing. The more accurate bottom-up inventories the authors cites (e.g., Jeong et al., 2014) can now identify hotspots with a much finer resolution. At the global scale, the source regions in this study may be considered hotspots, but those areas are really regions or subregions as shown in many previous studies.

Author's reply:
A hotspot does not have to be a very small area. It can be a large area, e.g., a country, see Oxford dictionary (https://en.oxforddictionaries.com/definition/hotspot): Definition: "A place of significant activity, danger, or violence." Example sentence: "Madagascar is considered a biodiversity hot spot, an area that is home to great numbers of species and that is under constant assault from human activity".
As we apply our method to areas of very different size and to areas emitting large amounts of methane, the term hotspot seems appropriate for this manuscript.

Q2: Referee:
Second, the authors try to match their satellite-based XCH4 to another assimilated product. This is disappointing because the value of those satellite products for XCH4 is significantly diminished as they are supposed to be used as independent retrievals of XCH4. The authors need a clear justification for this. Please see the related specific comments below.

Author's reply:
In the revised version of the paper we will improve the description of how the methane data product used for "matching" has been generated. This product uses optimized emissions (obtained via assimilation) which are then used to generate the atmospheric methane concentration (via forward simulation). Therefore, the atmospheric concentrations are consistent with the emissions and this is exactly what we need for our purpose. The correctness of the emissions is not relevant for our application but what is relevant is that the link between emissions and concentrations is modelled as good as possible.

Q3: Referee:
Third, it looks like that the proposed method ends up with a simple linear scaling of satellite-derived XCH4 to CAMS, in particular with a single parameter of V, which seems to be estimated as one value for the whole globe (as written it sounds like that; if not please clarify it).

Author's reply:
Yes, this understanding is correct. Please see also our detailed answer to the concerns of the other referee. In the revised version of our manuscript we will explain this better and will also add more details on our efforts to use meteorological data to improve on this. We also present an additional investigation using another model, which simulates methane at much higher spatial resolution compared to the used CAMS data set.

Q4: Referee:
Also, given the too large uncertainty for individual annual emission estimates, I wonder what value from this study can be added to the scientific community for regional GHG modeling.

Author's reply:
The purpose of our method is not to improve regional GHG modelling but to obtain very quickly (rough) methane emission estimates from (large amounts of) satellite data. The results can be used to identify regions where methane emissions are potentially higher than existing emission data bases suggest. We write in the "Summary and conclusions" section: "More detailed assessments likely require the use of much more complex approaches

compared to the simple method uses in this study. Nevertheless, simple and fast approaches also have a role to play as they permit to perform quick assessments on possible discrepancies with respect to emission inventories or other data sets and can also be used for plausibility checks for more complex approaches".

Q5: Referee:
Page 4, Lines 26 - 28, the sentence needs to be revised because the authors are trying to say two conflicting things in the sentence, making it confusing. Also, I would recommend that the authors be more quantitative instead of saying "agree reasonably well". In terms of data gap, how SCIAMACHY and GOSAT are different, e.g., available data points/pixels at the annual scale?

Author's reply:
It is not entirely clear for us why this sentence is confusing. Taking into account the sampling of GOSAT and the fewer number of observations (a factor of 2-3 depending on product) compared to SCIAMACHY we were also surprised about the reasonably good agreement of the emissions as obtained from SCIAMACHY and GOSAT. At present this is a finding based on our end results. We have not aimed at explaining this is in detail in terms of number of observations and required precision, accuracy and sampling as this is a complex topic requiring additional assumptions, e.g., on error correlations, and because we think that this a bit out of scope and not mandatory for our study. Concerning "agree reasonably well": We have added more specific information in the revised version of the manuscript by adding in brackets: "(e.g., in terms of mean value and scatter of the resulting annual emission estimates)". The difference in terms of data gaps is addressed in our manuscript as we show for each investigated target region $XCH_4$ maps for SCIAMACHY and GOSAT in Sect. 4.

Q6: Referee:
Page 5, Line 10, I wonder if the authors considered the data scarcity (i.e., small number of data) for the annual averages in terms of uncertainty. For certain pixels, the # of available data would be too small while others have enough for averaging.

Author's reply:
We consider this by visual inspection of annual $XCH_4$ maps for each target region (examples are shown in Sect. 4 of our manuscript) and quasi-automatically by varying the size and shape of the surrounding region and by considering the standard deviation of the resulting emissions in our error estimate. We are confident that this is better than explicitly using the number of individual data points for our error estimate as this would require knowledge on error correlations (please note that from previous studies we know that improvement upon averaging will not follow a square root law).

Q7: Referee:
Page 5, Line 17, I wouldn't use the term "enhancement" because the surrounding region is not equal to the CH4 "background" region, e.g., the Pacific region for the western US.

Author's reply:
In that paragraph we are not using the term "background". "Enhancement" is defined as source region $XCH_4$ minus surrounding region $XCH_4$. If this difference is positive than we have a positive enhancement, i.e., $XCH_4$ is higher over the source region compared to its surrounding area. In this context, it does not matter if the surrounding is equal to a true background or not. This only matters in terms of the accuracy of our method (e.g., additional sources in the surrounding region). In the revised version of our manuscript we have added more information on this accuracy aspect.

Q8: Referee:
Page 6, Lines 4 - 9, Looking at Eq. (1), the authors are trying to estimate emissions (flux) for the source region using _XCH4. But _XCH4 is not exactly the local enhancement, but only the relative enhancement to the surrounding region, which itself has some local enhancements. This will lead to underestimation of the emissions for the source region.

Author's reply:
Yes, this is true and in the revised version of our manuscript we will highlight this aspect more prominently and provide more details.

Q9: Referee:
Page 7, Lines 9 - 10, The authors confirm my point about the underestimation when using Eq. (1). The authors state that "we aim at quantifying the impact of the choice of the surrounding region by varying its size and shape."

This makes it very hard to adopt the proposed method in other regions because it involves adjustments of size and shape, likely yielding multiple estimates and subsequently expanding the uncertainty.

Author's reply:
Yes, this expands the uncertainty as explained in our paper. As our emission result depends on the chosen surrounding region our uncertainty estimate contains an error term which reflects this. Please note that it is not hard to adopt our method to other regions. For the four source areas discussed in our manuscript we vary the surrounding region using a pre-defined automatic procedure which is the same for all four source regions (see page 10, top).

Q10: Referee:
Page 7, Lines 20 - 22, There are two important concerns about the method. First, I expected from the title that the satellite products would provide independent observations as in most of the top-down studies. It is not vey satisfactory to try to match estimates from another product, i.e., CAMS. Also, from what is written here, I find that a single value for V needs a serious justification. Also, I am not convinced why CAMS should provide "true" estimates. Can the CAMS estimates be truly representative of any of the study sites/regions? How well are they compared with the estimates from previous studies for those source regions (maybe the word "true" may not be appropriate here; otherwise needs clarification).

Author's reply:
In the revised paper we will show additional results using another model which provides methane simulations at much higher spatial resolution. We also provide more details on why we are using a single value of V.
If we apply our method to real satellite data, then the true emissions are not known. However, if we apply our method to simulations the underlying emissions are known. We refer to these emissions as "true emissions", meaning "known emissions". In the revised version of our manuscript we will explain this better.

Q11: Referee:
With respect to the optimization of V, this parameter optimization would be the key to this study. However, it seems that there is no explanation or consideration of the errors between the relationship between CAMS and XCH4, which can be defined as: CAMS = f(XCH4, V) + err, where the function f is likely a linear one and err is the irreducible error (e.g., mean 0, normal error). Here for correct estimation of V, we need some independent estimates for err, similar to a linear regression case with errors.

Author's reply:
It is true that parameter V is very important as the estimated emissions are directly proportional to it. It is also true that it would be good to have an independent assessment of the error of our estimated emissions. Therefore, we have added in the revised version of our manuscript additional assessment results using another model to compute emission biases for several source regions and we use the results to present more details on the performance of our method.

Q12: Referee:
Page 10, Lines 18 - 21, I differ with the authors. The too large uncertainty suggests that the method is not powerful. I would conclude that the only value of the satellite products used in this study is to provide auxiliary information derived from the columnaveraged XCH4 which is linearly scaled to match another model product (rather than independent measurements).

Author's reply:
It is not clear for us what is wrong with what we write in lines 18 – 21. Our emission estimates are independent as they are derived from independent satellite retrievals. However, we agree that our large uncertainty limits the power of our method. In this context please see our response given above related to Q4.

Q13. Referee:
Page 11, 33-34, Again, the uncertainty is too large. When we think about hotspots, we expect relatively unambiguous isolation of emissions. The papers cited in this work already estimated emissions for the region with much better uncertainty. What policy makers need is identification of hotspots at the level of km scales and emission estimates for those small regions to mitigate sources from them. However, in this study, even the regional annual total yields very large uncertainty. Is there any way to reduce the uncertainty, even at the annual scale?

Author's reply:

Please see our response to your concern for Q12 and Q4. Our method is not accurate enough for "policy applications". This would require a much more powerful method. As explained above (and in our manuscript) the main purpose of our fast method is to obtain rough estimates of emissions for source regions of interest using large amounts of satellite data. Via our method, source regions can be identified where emissions are potentially significantly underestimated in emission inventories. These regions can then be studied in detail using more powerful (but also computationally much more demanding) procedures.

Q14: Referee:
Table 3. EDGAR v4.2 happens to estimate the same Mt CH4 for both Four Corners and the Central Valley?

Author's reply:
We have checked this for both source regions and found that the correct value for the Central Valley is 0.19, not 0.17. Many thanks for pointing to this! We have corrected Tab. 3.

Q15: Referee:
Figure 1. The region needs to be defined more accurately. For example, the region defined as the Central Valley of California in Figure 1 includes Southern California, and is different from that in Table 2.

Author's reply:
The purpose of Fig. 1 is to present an overview about the entire globe and to show where the investigated source regions are located and how XCH4 looks like in these areas but also in their surrounding area. It is not the purpose of Fig. 1 to define exactly the source regions. The exact definitions of the source regions is given in Tab. 2.

Q16: Referee:
Figure 8 needs some improvements. First, the data points (circles) should match the years on the X-axis label that are represented. Is the "standard deviation" the standard deviation of 7 annual estimates, e.g., for the 2003 - 2009. If this is the case, standard deviation is not very useful. I would be more interested in knowing the overall mean estimate for the multi-year period and the uncertainty about the mean, e.g., during 2003 - 2009. When individual annual estimates have huge uncertainties associated, I don't see the benefit of using standard deviation.

Author's reply:
We have improved this figure by changing the annotation of the x-axis (Year -> Time[year]). We use standard deviation as this is a precisely defined quantity in contrast to the computation of the uncertainty about the mean as this would require sufficiently good knowledge of error correlations.

Q17: Referee:
Also, the 1-sigma uncertainty in estimated emissions for individual years overlap with the EDGAR estimate, making it hard to statistically evaluate EDGAR. Looking at this at face value, I am not sure if there is any statistical power in the proposed method to say about the regional emission, even at the annual scale.

Author's reply:
Please see our response to these aspects (large uncertainty, power of our method) as given above.

[revised manuscript text omitted]

Several important issues for the future management and mitigation of methane emissions are not yet resolved  adequately, e.g., the methods  to verify  emission inventories and reported emissions per region (country down to city scale) (e.g., Ciais et al., 2014). The latter aspect  was studied in the development of the CarbonSat mission (Bovensmann et al., 2010; Velazco et al., 2011; Buchwitz et al., 2013; Pillai et al., 2016) for CO$_2$ using performance assessments based on simulated satellite observations (ESA, 2015) but so far only few studies have been published using real satellite data (e.g., Wecht et al., 2014a; Turner et al., 2015, 2016, for USA methane emissions). In this study we report an approach to use satellite methane retrievals to estimate  methane emissions of the two countries Azerbaijan and Turkmenistan, which are both important oil and gas producing countries, and also apply our method to two regions in the USA.

This manuscript is structured as follows: In Sect. 2 we introduce briefly the satellite data which have been used in this study. In Sect. 3 we describe the analysis method developed to derive methane emissions of (relatively) well localized areas

 from time-averaged satellite XCH$_4$ retrievals. The results as obtained from the satellite retrievals are presented and discussed in Sect. 4 and a summary and conclusions are given in Sect. 5.

**2 Satellite data**

During recent years the retrieval of near-surface-sensitive column-averaged dry-air mole fractions of atmospheric methane (CH$_4$) and carbon dioxide (CO$_2$), i.e., XCH$_4$ and XCO$_2$, from the satellite sensors SCIAMACHY (Burrows et al., 1995; Bovensmann et al., 1999) onboard ENVISAT and TANSO-FTS onboard GOSAT (Kuze et al., 2009, 2016) significantly

10    evolved and improved (e.g., Buchwitz et al., 2015, 2016a, 2016b; Butz et al., 2011; Dils et al., 2014; Frankenberg et al., 2011; Parker et al., 2011, 2015; Schneising et al., 2011, 2012, 2014; Yoshida et al., 2013).

For this study we use the latest data sets of XCH$_4$ retrievals from SCIAMACHY and GOSAT as generated by different research teams of the GHG-CCI project (Buchwitz et al., 2015) of ESA's Climate Change Initiative (CCI, Hollmann et al.,

15    2013). The four satellite XCH$_4$ products used for this study are publicly available and have been obtained from the GHG-CCI website (http://www.esa-ghg-cci.org; "latest data sets" refers to data access mid 2016; new versions are in preparation and are planned to be released in March 2017), 
[revised manuscript text omitted]

30 enhancement shown in Fig. 3 (a)).

$\Delta XCH_4$ also depends on the (size and shape) of the surrounding region.

 As explained below, we aim at quantifying the impact of the choice of the surrounding region by varying its size and shape.

Our method (Eqs. (1) and (2)) assumes a homogeneous distribution of emission sources ("flat source") within the chosen source region (Fig. 3). However, one would expect that due to atmospheric transport (advection and mixing) the observed atmospheric methane (e.g., for annual averages) typically covers a larger area than the underlying emission region(s). As can be concluded from Eqs. (1) and (2) our method results in an underestimation of the emissions, when this assumption is not valid. This can be seen as follows: Let's start with a situation, where our assumption is valid, i.e., there is a single homogeneous emission source region and its area is identical with the source region used for our analysis. In this case we obtain a certain value for $\Delta XCH_4$ and convert it to an estimated emission $E_e$ using conversion factor CF. Now let's assume that the surrounding area does not contain any emission sources. If we now extend the size of the source region (region A in Fig. 3) but do not change the outer boundary of the surrounding region (region B in Fig. 3), the true emission of the extended source region would be the same as before (as no emission sources are added, when the source region is extended) but the resulting methane enhancement ($\Delta XCH_4$) will decrease as the atmospheric methane enhancement will typically be the smaller the larger the distance from the source is. A smaller $\Delta XCH_4$ will result in a smaller value of the estimated emission, $E_e$ (see Eq. (1)). Conversion factor CF increases with increasing source region, i.e., the estimated emission not only depends on $\Delta XCH_4$ but also on the size of the source region via CF. The problem is that the increase of CF is only proportional to L, i.e., to the square root of the source area, whereas the decrease of $\Delta XCH_4$ may be proportional to the source area ($= L^2$). As a result, one would expect an underestimation of the estimated emission. This underestimation increases (gets worse) the more inhomogeneous the true emission sources are distributed within the investigated source region (an illustration is given below when discussing Figs. 9 and 10).

The value of V has been obtained by "calibrating" our method using global methane data sets obtained from the Copernicus Atmosphere Monitoring Service (CAMS, https://atmosphere.copernicus.eu/). Specifically, we use CAMS *a posteriori* methane emissions and corresponding atmospheric methane version v10-S1NOAA as generated via the TM5-4DVAR assimilation system assimilating National Oceanic and Atmospheric Administration (NOAA) $CH_4$ surface observations (an earlier version of this method and resulting data products is described in Bergamaschi et al., 2009). The CAMS data set used is based on forward modelling for the computation of atmospheric methane based on prescribed (but optimized) emissions. This is important as the calibration of our method requires atmospheric methane consistent with the underlying methane emissions. Based on this data set we computed annual emissions and corresponding annual $XCH_4$ at the original CAMS data set resolution of 6° longitude times 4° latitude. The corresponding maps for the year 2003 are shown in Fig. 4 (top row).

The CAMS year 2003 $XCH_4$ map shown in Fig. 4 top left has been used to derive methane emissions using Eq. (1) and varying parameter V (the only free parameter of our model) until the mean difference between our estimated emissions and

the "true" CAMS emissions is zero. We found that this is the case for V = 1.1 m/s (converted to km/year). The term "true" as used here (and below) does not imply that the CAMS emissions are perfect, i.e., free of errors. It simply means that these are the emissions which correspond to the atmospheric methane we use to calibrate our method, i.e., the atmospheric concentrations are computed using these emissions. What matters for our application is that we have a "good enough" modelling of the relationship between emissions and resulting atmospheric concentrations.

We found that this is the case for V = 1.1 m/s (converted to km/year). The resulting map of retrieved emissions using V = 1.1 m/s is shown in Fig. 43 bottom right. This map has been obtained using an automatic procedure: For all CAMS 6°x4° grid cells (except for the ones at the border) the $XCH_4$ value of this grid cell has been obtained and is interpreted as a potential source region value. The neighboring cells define the surrounding (background) of the potential source region and its $XCH_4$ mean value and standard deviation has been computed. A methane enhancement, $\Delta XCH_4$, has been computed as "source minus background value" (here "background" refers to the mean $XCH_4$ value in the surrounding region) as described above. If the resulting $\Delta XCH_4$ value is larger than 0.5 times the standard deviation of the $XCH_4$ values in the surrounding, then the corresponding cell is flagged as a methane "hotspot cell" and its $\Delta XCH_4$ value is converted to an emission using the approach described above (Eq. (1)). The corresponding results are shown as map in Fig. 43 bottom right and can be compared with the "true" emission map shown in Fig. 43 top right. As can be seen in Fig. 43, N = 125 hotspot cells have been found using the described procedure.

Figure 43 bottom left shows x-y plots of estimated emissions versus "true" (i.e., CAMS) emissions (top) and estimated minus true emissions versus true emissions (bottom). The mean difference "estimated-true" is 0.00 MtCH$_4$/year (this must be the case as V = 1.1 m/s has been determined by minimizing this difference). The standard deviation of the difference is 0.59 MtCH$_4$/year, the linear correlation coefficient R is 0.81 and the red line shows the resulting line from a linear fit. As can be seen, the (red) line originating from the linear fit has a positive slope but does not perfectly agree with the (green) 1:1 line (our single parameter model does not permit to also optimize the slope of the fitted line).

Figure 54 is similar as Fig. 43 but shows results for the year 2012. Here the difference "estimated-true" is not exactly zero but 0.01 MtCH$_4$/year. In contrast to Fig. 43, V has not been fitted. Instead, the pre-defined value of V = 1.1 m/s has been used. Figure 54 shows very similar "estimated-true" differences compared to Fig. 43. This indicatesdemonstrates that the effective wind speed V as obtained from year 2003 data is valid also for other years.

The results shown in Figs. 43 and 54 are combined in the single Fig. 65. As can be seen from Fig. 65 (top), the overall correlation of the retrieved and true emissions is 0.81, the mean difference (estimated minus true) is 0.00 MtCH$_4$/year and the standard deviation of the difference is 0.53 MtCH$_4$/year. As explained, these results have been obtained using constant values for wind speed fit parameter V (= 1.1 m/s) and correction factor C (= 2.0) (Eq. (2)). Several attempts have been

undertaken in order to find out if the use of regionally and/or time dependent V or C values can reduce the difference of the estimated and the true methane emission, however (so far) without success. For example, it has been investigated if the emission difference is correlated with mean wind speed (using ECMWF ERA Interim data obtained from www.ecmwf.int/, Dee et al., 2011) but no significant correlation between emission error and spatially resolved annual mean wind has been
5   found. Figure 7 illustrates this using annual mean wind speed at 900 hPa. As can be seen, there is essentially no correlation between emission error and mean wind speed (R = 0.049). Similar results have been obtained for other pressure levels (e.g., R = -0.036 for 800 hPa and R = 0.254 for the lowest ECMWF ERA Interim model level). This indicates that the use of mean wind speed (from meteorological data) does not help to improve the accuracy of our method. Future studies will show to what extent our method can be improved (or not). The year-to-year variation of the estimated annual
10   emission, $E_e$, for a given satellite $XCH_4$ product is therefore  entirely driven by the satellite-derived methane enhancement, $\Delta XCH_4$, as parameters V and C are constant.

Finally, the (1-sigma) uncertainty of $E_e$ has been estimated. This has been done as follows: Figure 6 also shows the emission difference ("estimated minus true"; see middle and bottom panels) as a function of the estimated emission. Figure
15   6 middle also shows (in red) the corresponding mean values (crosses) and standard deviations (vertical bars) for several emission bins (non-equidistant to ensure a sufficiently large number of data points within each bin). Also shown in Fig. 6 (middle and bottom) are dotted red lines computed as $f(E_e) = 0.3 + 0.5 \cdot E_e$. This function and its parameters has been chosen such that the red vertical bars (1-sigma range) are located within the range defined by $f(E_e)$, i.e., most of the emission differences are located within +/- $f(E_e)$  (Fig. 6 middle). Therefore, $f(E_e)$ is a reasonable description of the 1-sigma
20   uncertainty of the estimated emissions. Based on this it is concluded that the 1-sigma uncertainty of the estimated emission due to uncertainty of the overall conversion factor (CF) can be well described using this formula:

$$\sigma_{CF} = 0.3 + 0.5 \cdot E_e. \hspace{5cm} (4)$$

25   Here the units of $\sigma_{CF}$ and $E_e$ are MtCH4/year. The total uncertainty, $\sigma_{tot}$, consists of the uncertainty of the conversion factor, $\sigma_{CF}$, and the uncertainty of the obtained methane enhancement, $\sigma_{\Delta XCH4}$, as obtained from the satellite data (see Eq. (1)). The latter is assumed to be dominated by methane variations in the surrounding area (primarily because the surrounding region may contain regions of elevated methane due to sources located outside the source region). This contribution to the total uncertainty is estimated by varying the size of the surrounding region (see following section). The total uncertainty is
30   computed as follows:

$$\sigma_{tot} = \sqrt{\sigma_{\Delta XCH4}{}^2 + \sigma_{CF}{}^2} \hspace{5cm} (5)$$

The method described in this section has been applied to the described SCIAMACHY and GOSAT XCH$_4$ data products and for each of the pre-defined source regions annual average emissions and their uncertainties have been obtained for all products. The results are presented in Sect. 4. Before the method is applied to real data it is relevant to carry out some additional investigations using simulations as in this case the "true emissions" are known. For this purpose, a high-resolution methane data set is used to investigate how well the inversion method performs when using a different model, which simulates atmospheric methane at much higher spatial resolution than the model described and used for the results presented in this section. The high-resolution results are presented in the following sub-section 3.1.

**3.1 Performance of inversion method as applied to simulations of high-resolution methane**

In order to test the inversion method using a methane data set at higher resolution, simulated atmospheric methane concentrations using posterior methane emissions from Turner et al., 2015, have been used. The spatial resolution of this data set is 0.5° latitude times 0.667° longitude and it covers North America. The methane concentrations have been computed with GEOS-Chem. This data set is referred to as GCT15 in this manuscript. It covers one year (2010) and consists of methane emissions and corresponding atmospheric concentrations on the same spatial grid.

Figure 8 shows (around noon) annually averaged GCT15 XCH$_4$ over the USA. As can be seen, there are several regions, where methane is significantly enhanced compared to their surrounding areas. However, one would see even more "emission hotspot areas", when zooming into this map and when using an appropriate color scale for the zoomed-in regions.

This is demonstrated in Fig. 9a focusing on central California (a region discussed in detail in Sect. 4). As can be seen, there is a region of clearly elevated methane (red color) located approximately between the two cities Modesto and Merced (not shown). This region has been selected as a source region shown as polygon (thick black line) in Fig. 9a and is referred to as "California(MM)" (CMM) in the following. The "surrounding region" as used to compute $\Delta$XCH$_4$ (via "source – background" XCH$_4$) is shown as white rectangle. As shown in Fig. 9, $\Delta$XCH$_4$ is 9.4 ppb, and the estimated emission of the CMM region, computed using Eq. (1) with the parameters described earlier, is 729 +/- 664 ktCH$_4$/yr. The GCT15 emissions, i.e., the "true" emissions, are shown in Fig. 9b and the emission is 727 ktCH$_4$/yr in the CMM source region. It needs to be pointed out that the GCT15 emissions can be large outside the selected CMM source region, in particular in the San Francisco area (the red cell corresponds to an emission of nearly 200 ktCH$_4$/yr) but this major source region is located outside the selected source region, which is defined based (only) on XCH$_4$ (Fig. 9a). The excellent agreement of the estimated emission and the true emission can, of course, be simply by chance in this case. Here it is likely that XCH$_4$ over the CMM region is (due to transport) significantly affected by San Francisco emissions, i.e., by emission located outside the source region (see also Bao et al., 2008, for a discussion of the meteorology in this area). Therefore, one has to be careful

when interpreting the estimated emissions as they may also be influenced by emission sources in the surroundings. On the other hand, there is also outflow from the source region into the surrounding region. All this (and other aspects) result in quite large uncertainty of the estimated emission and this is reflected in the uncertainty estimate, which is quite conservative, i.e., it is quite large. In this case, our estimated (1-sigma) uncertainty is 664 $ktCH_4$/yr, which is nearly 100% of the estimated emission. This uncertainty has been computed for the surrounding region shown in Fig. 9, i.e., by neglecting the additional error contribution due to variations of the surrounding region ($\sigma_{\Delta XCH4}$ in Eq. (5)). This contribution is however small compared to error term $\sigma_{CF}$ (= 664 $ktCH_4$/yr in this case). That the total uncertainty is typically clearly dominated by $\sigma_{CF}$ is a finding that has also been confirmed when analyzing the real satellite data (see Sect. 4), where both uncertainty contributions are always considered.

Figure 10 shows similar results to those in Fig. 9 but for an extended source region, denoted CMS in the following. This region covers the region from near San Francisco in the north to Los Angeles in the south. As can be seen, $\Delta XCH_4$ is 7.2 ppb and the estimated emission is 770 +/- 685 $ktCH_4$/yr, which is significantly lower than the "true" CMS region emission of 1228 $ktCH_4$/yr, i.e., in this case the estimated emission is wrong by -37% (computed as "(estimated – true)/true"). However, the true emission is inside the uncertainty range of the 1-sigma range of the estimated emission (but close to the upper edge of the uncertainty range, which is 1455 $ktCH_4$/yr). The reason for this underestimation is very likely due to the fact that the emission sources are distributed very irregularly inside the CMS region. As already explained above, a significant underestimation of the estimated emission is expected in this case.

As can also be seen from Fig. 10a, there is a region of clearly elevated $XCH_4$ in the southern part of the CMS source region. This region corresponds to the Los Angeles area. Figure 11a shows a zoom into this region. In this case we define the source region by a simple rectangle. The estimated Los Angeles area methane emission is 250 +/- 425 $ktCH_4$/yr, whereas the true emission is 367 $ktCH_4$/yr, i.e., the difference -32% (negative, i.e., the estimated emission is (again) underestimated).

Another interesting source region is the Four Corners, which is discussed in detail in Sect. 4. As shown in Fig. 12, the estimated emission is 795 +/- 697 $ktCH_4$/yr, whereas the "true emission" is 1404 $ktCH_4$/yr, i.e., the difference -43%.

Comparisons of estimated versus true emissions such as those presented here have also been carried out for several other of the methane emission hot spot area shown in Fig. 8. Figure 13 presents an overview of the corresponding results. As can be seen, the estimated emissions are typically underestimated by about 40%. The emission uncertainties are large (on the order of 100%) but the true emissions are within the 1-sigma uncertainty estimate of the estimated emission (with one exception: Chicago area: here the true emission is 1473 $ktCH_4$/yr but the upper (1-sigma) range of the estimated emission is 1322 $ktCH_4$/yr). Based on these results it is concluded that the estimation method as described in this manuscript provides reasonable results but with a clear tendency to underestimate the emissions (as expected from the theoretical considerations

presented earlier). To what extent the 40% value depends on the model used (in this case GEOS-Chem) and on its characteristics (such as spatio-temporal resolution) needs to be investigated (e.g., by using also other models). In any case, the results presented in this section need to be considered when interpreting results obtained from applying this method to real satellite $XCH_4$ retrievals as 
[revised manuscript text omitted]

The inversion method has been tested by applying it to a high-resolution methane data set covering the USA, which has been computed with GEOS-Chem. We retrieve methane emissions for several areas where the GEOS-Chem data set shows elevated $XCH_4$ compared to their surrounding areas. We found that the estimated emissions are typically 40% lower compared to the emissions used in the model (which are the known, i.e., "true" emissions of this simulation experiment). The true emissions are (with one exception) located within the 1-sigma uncertainty range of our emission estimates. From theoretical considerations we expect that our method tends to underestimate emissions, i.e., that it provides rather conservative emission estimates. To what extent the 40% value depends on the model used and on its characteristics (such as spatio-temporal resolution) needs to be investigated in the future by using additional models.

[revised manuscript text omitted]

**Figure 3A1.** Sketch of a simple model used to explain the methane emission estimation method described in Sect. 3. (a) Source region A (of size $L_xL_y$ and with $L_x$ in wind speed direction (wind speed magnitude V)) with elevated $XCH_4$ (light red) and surrounding (background) region B (white area). (b) Air parcels (blue squares) moving with constant speed V over a source region with emission $E/(L_xL_y)$, where E is the source area emission in $CH_4$ mass per time, while accumulating methane during accumulation time $\tau$ (= $L_x/V$). (c) Before entering the source region, the air parcels are characterized by a background methane vertical column, $VC_b$, in units of $CH_4$ mass per area. When leaving the source area their vertical column has been enhanced by $\Delta VC = E/(L_xL_y) \cdot \tau$. When passing over the source region, their vertical column increases linearly and, therefore, the average column enhancement over the source region is $0.5 \cdot \Delta VC$. VC ($CH_4$ mass per area) can be converted to $XCH_4$ (ppb) via a factor M (in units of mass and per area and per ppb).

[Figure]

**Figure 43.** Methane emissions (in MtCH_4/year) and corresponding XCH_4 (in ppb) for the year 2003 at 6° longitude times 4° latitude resolution. Top left: XCH_4 as computed from Copernicus Atmosphere Monitoring Service (CAMS) atmospheric CH_4 fields (version v10-S1NOAA; resolution: 6°x4°; obtained from https://atmosphere.copernicus.eu/). Top right: Corresponding CAMS total, i.e., anthropogenic and natural, methane emissions. Map bottom right: Methane emissions of (automatically determined potential) emission hot spots ("hotspot cells") as derived from the top left XCH_4 map using the method described in Sect. 3. Bottom left: Comparison of retrieved emissions (map bottom right) with the "true" CAMS emissions (map top right). Here N (= 125) denotes the number of grid cells for which emission values have been obtained ("hotspot cells", see main text for details), R (= 0.81) is the linear correlation coefficient of retrieved and true emissions, and D is the difference between the retrieved and the true emissions in terms of mean difference and standard deviation (0.00 +/- 0.59 MtCH_4/year).

[Figure]

**Figure 54.** As Fig. 43 but for year 2012.

[Figure]

**Figure 65.** Top: "True" (i.e., CAMS) emission, $E_t$, versus estimated emissions, $E_e$, as obtained from the simulation-based

5    assessment results shown in Figs. 43 and 54 (i.e., shown are all "hotspot cells" also shown in these two figures, see caption

Fig. 43 and main text for details). Middle and bottom: Emission difference "estimated minus true" versus estimated

emission. The grey vertical bars denote the boundaries of emission bins for which mean differences (red crosses) and

standard deviations of the differences (red vertical lines) have been computed. The red dotted line shows that the relationship

between the estimated emission ($E_e$) and its 1-sigma uncertainty ($\sigma$) can be approximately described by $\sigma(E_e) = 0.3 + 0.5\,E_e$.

[Figure]

**Figure 7.** Error of the estimated emission (black symbols; computed as "retrieved – true", see Fig. 6) versus annual mean wind speed (red crosses) at 900 hPa. Top: all data; bottom: same data but x-y zoom. The linear correlation coefficient between annual emission error and annual mean wind speed is 0.049.

[Figure]

**Figure 8.** Anually averaged (year 2010) atmospheric column-averaged methane (XCH₄) computed with GEOS-Chem using *a posteriori* methane emissions of Turner et al., 2015 ("GCT15 data set"). The resolution of this data set is 0.5° latitude x 0.667° longitude.

[Figure]

**Figure 9.** Top: GCT15 XCH$_4$ over parts of California. The white rectangle denotes the "surrounding region" of the "source region", which is surrounded by a polygon shown as thick black line. The source region covers the area between the two cities Modesto and Merced in central California. The text below lists the XCH$_4$ enhancement, $\Delta$XCH$_4$ (9.4 ppb) and the estimated emission (729 +/- 664 ktCH$_4$/yr). The "true" emission of the source region has been computed from the GCT15 emissions (bottom panel) and is 727 ktCH$_4$/yr.

[Figure]

(a) Methane GCT15 / California(MS) - 2010

XCH$_4$ [ppb]

1752    1759    1766    1773    1780

XCH$_4$ enhancement [ppb]:   7.2

Estimated CH$_4$ emission [ktCH$_4$/yr]: 770.0 +/- 685.0

True CH$_4$ emission [ktCH$_4$/yr]:      1228.0

(b)    Methane emission GCT15

CH$_4$ emission [ktCH$_4$/yr] (0.667x0.5)

0       45      90      135     180

**Figure 10.** As Fig. 9 but for a larger part of California, referred to as "California Mid/South (MS)" in this publication.

[Figure]

(a) Methane GCT15 / Los Angeles - 2010

XCH$_4$ [ppb]

1760    1765    1770    1775    1780

XCH$_4$ enhancement [ppb]:    4.9

Estimated CH$_4$ emission [ktCH$_4$/yr]: 250.4 +/- 425.2

True CH$_4$ emission [ktCH$_4$/yr]:        366.5

(b)    Methane emission GCT15

CH$_4$ emission [ktCH$_4$/yr] (0.667x0.5)

0        35        70        105        140

**Figure 11.** As Fig. 9 but for the region around Los Angeles, California (this region is located in the southern part of the source region shown in Fig. 10).

[Figure]

[Figure]

XCH₄ enhancement [ppb]: 23.8

Estimated CH₄ emission [ktCH₄/yr]: 794.8 +/- 697.4

True CH₄ emission [ktCH₄/yr]: 1404.2

[Figure]

**Figure 12.** As Fig. 9 but for the Four Corners region (see main text for details).

[Figure]

**Figure 13.** Methane emission estimates for several methane hot spot areas in the USA as obtained by applying out simple mass balance method to the year 2010 GCT15 data set. The figure in the center is identical with Fig. 8 and shows year 2010 $XCH_4$ over the USA. For each hotspot region the following three numbers are listed below each map: Estimated methane and its uncertainty in $ktCH_4$/yr for the shown source regions (thick black lines, mostly rectangles). The number in brackets is the percentage difference of the estimated emission and the corresponding true emission (computed as "(estimated – true)/true"), where the true emission is the source region GCT15 emission.

[revised manuscript text omitted]

v4.2 (FT2012) annual emissions for Turkmenistan.

[Figure]

**Figure A1.**
5

10